

# Proposal for an autonomous quantum heat engine

**Miika Rasola[1⋆], Vasilii Vadimov[1†], Tuomas Uusnäkki[1] and Mikko Möttönen[1,2‡]**

**1** QCD Labs, QTF Centre of Excellence, Department of Applied Physics, Aalto University,
P.O. Box 13500, FI-00076 Aalto, Finland
**2** QTF Centre of Excellence, VTT Technical Research Centre of Finland Ltd.,
P.O. Box 1000, 02044 VTT, Finland

⋆ miika.rasola@aalto.fi , † vasilii.1.vadimov@aalto.fi , ‡ mikko.mottonen@aalto.fi

## Abstract

We propose and theoretically analyse a superconducting electric circuit which can be used to experimentally realize an autonomous quantum heat engine. Using a quasi-classical, non-Markovian theoretical model, we demonstrate that coherent microwave power generation can emerge solely from the heat flow through the circuit determined by non-linear circuit quantum electrodynamics. The predicted energy generation rate is sufficiently high for experimental observation with contemporary techniques, rendering this work a significant step toward the first experimental realization of an autonomous quantum heat engine based on Otto cycles.



## Contents


# 1   Introduction

Superconducting quantum circuits provide one of the most versatile platforms for experimental implementation of various quantum technological concepts and devices [1]. The field of quantum microwave engineering, serving a purpose beyond fundamental research with the goal of realizing practically useful quantum devices [1, 2], has already produced numerous groundbreaking results in quantum computation [3–7], communication [8–11], simulation [12–18], and sensing [19–26].

As quantum technology evolves further, the necessity for understanding thermodynamics at the quantum level becomes increasingly urgent. On the one hand, quantum thermodynamics [27–29] seeks to extend the theories of classical thermodynamics into the domain of single quantum systems in order to improve our understanding of microscopic and mesoscopic out-of-equilibrium systems with heat flows. On the other hand, the theoretical concept of a thermodynamic cycle, along with the heat engine that implements it, has been foundational to thermodynamics since its inception in the 19[th] century. Thus, it is not surprising that they remain indispensable in the quantum era.

Although microwave quantum engineering has led to significant breakthroughs in recent decades, only one experimental realization of a reciprocating quantum heat engine (QHE) in superconducting quantum circuits has been reported to date [30]. Another device capable of operating in the continuous QHE regime is reported in Ref. [31]. In contrast, numerous experimental demonstrations of QHEs have been achieved in microscopic atomic devices, such as single trapped ions [32], a spin coupled to single-ion motion [33, 34], nitrogen-vacancy centre interacting with a light field [35], nuclear magnetic resonance [36, 37], and large quasi-spin states of caesium impurities immersed in an ultracold rubidium bath [38]. However, a substantial amount of experimental studies about heat conduction and thermodynamics in superconducting circuits have been published, providing prospects for further advancement [39–44].

Autonomous quantum heat engines are a class of particularly interesting thermal devices operating at the level of a few excitation quanta and capable of autonomous operation, where they only utilize the flow of heat as a resource in order to provide useful work. All of the abovecited thermal machines are inherently non-autonomous because their thermodynamic cycles need to be persistently driven by some type of external control. In this case, it is very challenging to generate more work output than the energy consumed for the external control and to even extract the work produced by the heat engine, as it typically only exists superimposed on a macroscopic external control field.

Although superconducting quantum circuits have yet to flourish in the field of quantum thermal machines, we consider that they offer a promising platform for realizing the first autonomous QHE implementing a thermodynamic cycle. In this article, we build upon our previous work [45], where we proposed a general-level theoretical approach for realizing an autonomous QHE using an arbitrary system governed by an optomechanical Hamiltonian [46,47] and coupled to two narrow-band thermal reservoirs [48]. Here, we focus on an experimentally feasible superconducting circuit and derive an in-depth theoretical model for it, starting from the first principles, explicitly linking the circuit to the physical quantities defining its electrodynamics. Furthermore, we show that in a carefully constructed superconducting circuit, a resonator acting as the controller of the cycle can exhibit effectively negative internal dissipation, i.e., coherent microwave power generation, arising solely from the internal dynamics of the QHE and the corresponding heat flow through the circuit.

We stress that various theoretical proposals of QHEs, both in optomechanical systems and quantized superconducting circuits, from other authors precede this work. References [49–51] extensively analyse the possibility of realizing a coherently driven quantum Otto cycle in an optomechanical system, whereas Refs. [52–54] utilize optomechanical systems, but rely on periodic incoherent thermal drives. Hardal et al. [54] even propose a device based on superconducting circuits, but leave the discussion on a general level without specifying the circuit in detail. Furthermore, Refs. [55–57] study theoretical models of autonomous QHEs on the level of Markovian master equations. In contrast to these high-level theoretical proposals, we take the discussion closer to a readily designed device by analysing the circuit starting from the physical quantities defining the circuit. In addition, our proposed QHE is fully autonomous without any external persistent periodic thermal pumping or coherent drive. Finally, as opposed to the other approaches, our theoretical description is non-Markovian, taking into account the peaked shape of the reservoir spectra, which seems to be required to satisfactorily describe an autonomous QHE.

The article is organised as follows. In Sec 2, we define the circuit in detail, specifying the necessary physical quantities and parameters. We also outline the theoretical framework used to analyse the system, including the required approximations. In Sec. 3, we present a step-by-step derivation of the theoretical model, providing all relevant calculations before arriving at the final result. We then apply this model to a specific set of circuit parameters in Sec. 4, demonstrating that the circuit can operate as an autonomous quantum heat engine. By performing multiple parameter sweeps, we analyse the behaviour of the circuit and compare its dynamics to the quantum Otto cycle. Finally, in Sec. 5, we summarise our findings and discuss potential future research directions and applications for the proposed device.

## 2 Device

Before specifying the electric circuit to be analyzed, let us review the ideas presented in Ref. [45] to provide an intuitive description of the dynamics we are seeking with the proposed device. Consider the traditional quantum Otto cycle as presented by the schematic in Fig. 1(a). The engine consists of a working body and two heat baths with peaked spectral densities localized at angular frequencies $\omega_{\mathrm{h}}$ and $\omega_{\mathrm{c}}$ for the hot and cold baths, respectively. The characteristic angular frequency of the working-body mode is periodically tuned between the higher angular frequency, $\omega_{\mathrm{h}}$, and the lower angular frequency, $\omega_{\mathrm{c}}$. The coupling between the working body and its baths is constantly on; however, the energy exchange is the most efficient when the working-body mode is in resonance with a heat bath.

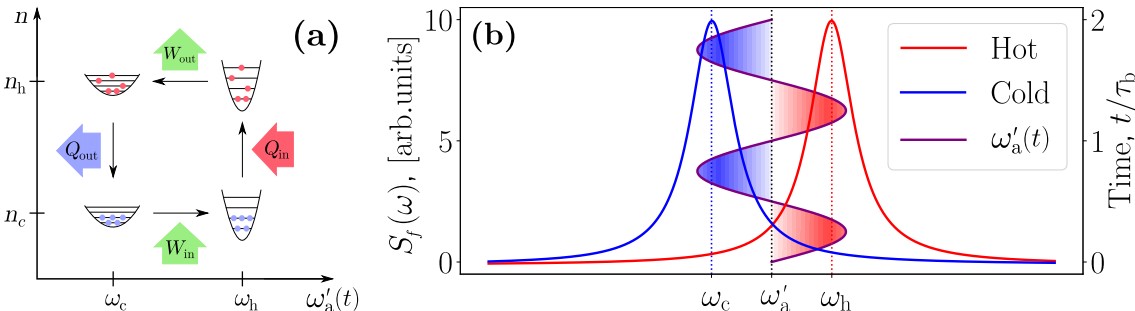

Figure 1: Schematics illustrating the dynamics of the proposed quantum heat engine. (a) Idealised description of the quantum Otto cycle. The working-body mode is depicted as a quadratic potential traversing between frequency values $\omega_c$ and $\omega_h$. In the photon number representation, the average photon numbers after interacting with the heat reservoirs are given by $n_c$ and $n_h$, respectively. The work done by the oscillator $W_{out}$ exceeds the work done on the oscillator $W_{in}$. Consistently, the heat obtained from the hot reservoir by the system $Q_h$ exceeds the heat released from the system to the cold reservoir $Q_c$. (b) Feasible Lorentzian line shapes of the cold (blue colour) and hot (red colour) reservoirs with indicated centre angular frequencies $\omega_c$ and $\omega_h$, respectively, along with the frequency-modulated angular frequency $\omega_a'(t)$, shown by the sinusoidal purple line. The shaded areas depict regions in time and frequency where the effective resonator interacts with the heat reservoirs, with darker colours depicting stronger coupling.

After resonant interaction with one of the heat reservoirs, the average photon number of the working-body mode is given by $n_h > n_c$, for hot and cold reservoirs, respectively. When the angular frequency of the working body is far away from resonance with either of the baths, its photon population is essentially conserved. Therefore, the working-body mode pumps photons from the high-frequency high-temperature environment to the low-frequency low-temperature environment, resulting in energy being released to the driving component owing to energy conservation.

As mentioned above, the driving component is typically an external macroscopic field, leading to the small power output of the quantum heat engine to be irreversibly lost into the macroscopic field. Here, we replace the external control field with a mesoscopic oscillator controlling the angular frequency modulation of the working body. The angular frequency of the driving mode, $\omega_b$, determines the modulation period, $\tau_b = 2\pi/\omega_b$, of the working-body angular frequency, $\omega_a'(t)$, as depicted in Fig. 1(b).

The heat reservoirs are realized by noise-driven filters with Lorentzian line shapes at their respective central frequencies, which naturally endow the device with a frequency-dependent coupling between the working body and the reservoirs, as depicted in Fig. 1(b). In this article, we show that it is possible to construct an electric circuit with the above-mentioned properties such that the working-body mode undergoes an approximate Otto cycle and the power generated by the heat engine can be extracted from the mode of the driving resonator.

In the context of a mechanical heat engine, the working-body mode exactly assimilates the working body of the engine, while the driving mode acts as a flywheel that facilitates work output. To provide intuition, one may also study an analogue to an internal combustion engine, where the working body resonator is considered as the piston of the combustion engine moving within the cylinder, and the driving resonator as the combination of the crankshaft and the flywheel.

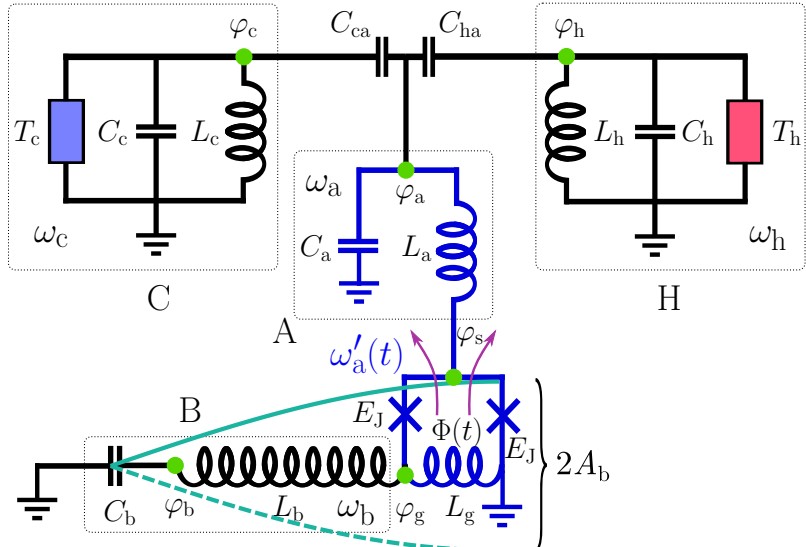

Figure 2: Lumped-element circuit model of the quantum heat engine. The heat engine circuit consists of four LC resonators denoted by A, B, C, and H as marked by the dotted line boxes. The main capacitances and inductances of the resonators are denoted by $C_i$ and $L_i$, respectively, with $i = a, b, c, h$. The bare angular frequencies of the resonators are given by $\omega_i = 1/\sqrt{L_i C_i}$. The capacitances $C_{ca}$ and $C_{ha}$ couple resonator A to resonators C and H, respectively. Resonators C and H contain dissipative elements, indicated by the blue and red boxes, maintained at temperatures $T_c$ and $T_h$, respectively. The Josephson junctions in the symmetric SQUID loop both have a Josephson energy of $E_J$. In addition, a small piece of an inductor directly connected to resonator B with the inductance $L_g$ forms a part of the SQUID loop, through which resonator B is also grounded. We indicate the circuit nodes with green dots, and assign a flux variable $\varphi_i$, with $i = a, b, c, h, s, g$, for each node. The flux-tunable effective resonator acting as the working body of the quantum heat engine is denoted by the blue colour in the centre, with the time-dependent and flux-tunable frequency $\omega_a'(t)$. The purple arrows illustrate the total magnetic flux $\Phi(t) = \Phi_{ext} + \varphi_g(t)$ of the SQUID loop, with $\Phi_{ext}$ as the static external magnetic flux. The turquoise quarter wavelength cosines illustrate the field mode $\varphi_b$ in resonator B.

## 2.1 The circuit

Next, we define the electric circuit with the resonators housing the working-body mode and the driving mode, as well as the heat reservoir filters. In Fig. 2, we present a lumped-element circuit model for the QHE and assign the required physical quantities. The circuit consists of four inductor-capacitor (LC) resonators denoted as A, B, C, and H, each comprising of a capacitance, $C_i$, and an inductance, $L_i$, with $i = a, b, c, h$. The bare angular frequencies of the resonators, denoted in Fig. 2, are given by $\omega_i = 1/\sqrt{L_i C_i}$. In the following, we use the subscripts "h" and "c" for "hot" and "cold", respectively. Resonators C and H, collectively referred to as the filters, contain dissipative elements at temperatures $T_c < T_h$, respectively, and act as the non-Markovian heat reservoirs with Lorentzian-shaped power spectral densities. The filters are linearly coupled to resonator A via the coupling capacitances $C_{ha}$ and $C_{ca}$. The core of the QHE is formed by the coupled system of resonators A and B, where the non-linear coupling implemented by a SQUID enables the cyclic dynamics of the heat engine. The inductor of resonator A is grounded through this symmetric SQUID, such that a segment of the inductor of resonator B becomes a part of the SQUID loop, establishing the non-linear coupling between resonators A and B.

To relate the above discussion about the working-body mode and driving mode to the circuit, we define the working-body resonator as the effective resonator consisting of the bare resonator A and the shunting SQUID that grounds resonator A, as depicted in Fig. 2. The angular frequency, $\omega'_a$, of this effective resonator can be tuned by the static external flux $\Phi_{ext}$, but it is also modulated by the oscillating magnetic flux in the coupling inductor $L_g$, $\varphi_g(t)$. This leads to the above-discussed dynamics where the angular frequency $\omega'_a(t)$ of the working body traverses, driven by the slow oscillations in resonator B, between the resonance frequencies of the heat reservoirs, as depicted in Fig. 1(b).

In addition to the physical structure and quantities, we define six flux node variables [58, 59], one for each resonator and two auxiliary node variables, $\varphi_s$ and $\varphi_g$, for the purpose of modelling the SQUID-mediated coupling correctly. First, we divide the field in resonator B into two components, but subsequently derive an effective-field approach unifying the two fields into a single equation of motion. The field $\varphi_g$ can be identified as the coupling field, i.e., the part of resonator B that couples to the field $\varphi_s$. Similarly, $\varphi_s$ represents a small deviation from $\varphi_a$, separated by the inductance $L_a$, that couples to $\varphi_g$ via the SQUID. In the following, we will rigorously derive the equations of motion for these fields and show that the mode in resonator B can exhibit coherent microwave power generation arising solely from the internal dynamics of the device involving the heat flow from the hot to the cold reservoir.

## 2.2  Initial assumptions and the recipe for solution

As the final step in our preparation, we set some initial assumptions about the parameters of the system necessary for achieving the desired QHE dynamics and outline the approach for modelling the system. The primary goal of this circuit is to achieve dynamics where resonator A undergoes a cycle analogous to the quantum Otto cycle [27,60] driven by the field amplitude of resonator B. This concept, along with most of the assumptions made here, has been thoroughly discussed phenomenologically in Ref. [45]. Therefore, we will simply list the assumptions here without delving into detailed reasoning.

The first requirement, as mentioned above, is the existence of heat reservoirs at different temperatures: $T_c < T_h$. We have already assigned angular frequencies for the filter resonators, operating as pass-band filters between the corresponding dissipative elements and resonator A, but necessarily, the hot filter must have a higher frequency than the cold filter. In Ref. [45], the following condition for the angular frequency of resonator A is set: in the operational mode, the mean angular frequency of the working body, $\omega'_a$, preferably lies between the filter frequencies, resulting in the condition $\omega_c \lesssim \omega'_a \lesssim \omega_h$. In the current circuit, $\omega'_a$ is an effective angular frequency of the subcircuit consisting of $C_a$, $L_a$, and the SQUID loop. By virtue of the SQUID, this frequency becomes flux-tunable and must be tuned to a suitable value by the static external flux $\Phi_{ext}$. Below, we find an approximate expression for $\omega'_a$ and show that, although the exact formulation of this condition differs slightly within the framework used here, its fundamental principle remains unchanged. For successful tuning, the frequency difference between the filter resonators desirably exceeds the capacitive coupling strength between the filters and resonator A arising from $C_{ca}$ and $C_{ha}$. Lastly, we impose the condition that the angular frequency of resonator B is significantly lower than that of the other resonators: $\omega_b \ll \omega_c, \omega_a, \omega_h$ [45].

Our goal is to theoretically show that the dissipative dynamics of the circuit can lead to observable coherent microwave power generation in resonator B. To this end, we identify three characteristic time scales in the system: the short time scale governed by the high frequencies $\omega_a$, $\omega_h$, and $\omega_c$, the intermediate scale associated with the angular frequency $\omega_b$, and the long time scale, determined by the rate at which the average occupation or field amplitude in resonator B changes. The rate of energy generation is proportional to the long time scale. We shall further assume that the rate of generation is considerably lower than the angular frequency

of resonator B, $\omega_b$, which effectively allows using the Wentzel–Kramers–Brillouin (WKB) approximation [61] for the resonator mode $\phi_b$. Based on this assumption, we systematically eliminate all other degrees of freedom except those of the mode in resonator B, ultimately arriving at an equation of motion for the amplitude and phase of the mode in resonator B.

In order to facilitate simultaneous modelling of both non-linear and non-Markovian properties of the circuit, we resort to a quasiclassical theoretical model. We model the noise spectral density by the quantum mechanical result, but treat the circuit degrees of freedom classically. Since a linear system retains the form of its equations of motion in quantization, and we assume the non-linear component to be a small perturbation to the linear dynamics, we expect that the quasiclassical model satisfactorily describes the system.

# 3 Theory

## 3.1 Equations of motion

Excluding the SQUID and the dissipative components, the classical Lagrangian for the circuit described in Fig. 2 is given by

$$\mathcal{L}_R = \frac{C_{\Sigma a}\dot{\varphi}_a^2}{2} - \frac{(\varphi_a - \varphi_s)^2}{2L_a} + \frac{C_b\dot{\varphi}_b^2}{2} - \frac{(\varphi_b - \varphi_g)^2}{2L_b} + \sum_{f=c,h}\left[\frac{C_{\Sigma f}\dot{\varphi}_f^2}{2} - \frac{\varphi_f^2}{2L_f} + C_{fa}\dot{\varphi}_a\dot{\varphi}_f\right], \quad (1)$$

where we define the capacitance sums $C_{\Sigma a} = C_a + C_{ha} + C_{ca}$ and $C_{\Sigma f} = C_f + C_{fa}$. Neglecting the capacitance of the Josephson junctions, the Lagrangian, including the elements in the SQUID loop providing the coupling between resonators A and B, is given by

$$\mathcal{L}_S = -\frac{\varphi_g^2}{2L_g} + E_J\cos\left(\frac{2\pi\varphi_s}{\Phi_0}\right) + E_J\cos\left[\frac{2\pi}{\Phi_0}\left(\varphi_s - \varphi_g - \Phi_{ext}\right)\right], \quad (2)$$

where $E_J = I_c\Phi_0/(2\pi)$ is the Josephson energy of the junctions with the critical current $I_c$ and the magnetic flux quantum $\Phi_0 = \pi\hbar/e$, defined by the reduced Planck constant, $\hbar$, and the elementary charge, $e$. The total Lagrangian of the circuit without any additional approximations is thus given by the sum $\mathcal{L} = \mathcal{L}_R + \mathcal{L}_S$.

Typically, the trigonometric potential in the SQUID Lagrangian is Taylor expanded up to the first non-linear correction. Before such treatment, we need to ensure that the field variables in the cosine are sufficiently small. To this end, we find the minimum of the potential energy related to the above SQUID loop Lagrangian (2), and define new shifted field variables as deviations from the potential minimum for the fields inductively coupled to the SQUID. The new field variables are given as $\tilde{\varphi}_i = \varphi_i - \varphi_g^{(0)}/2$, where $i \in [s, a, b, g]$, and $\varphi_g^{(0)}$ is the field at the potential minimum, given by the equation

$$\frac{\varphi_g^{(0)} - \Phi_{ext}}{L_g} + 2I_c\sin\left(\frac{\pi}{\Phi_0}\varphi_g^{(0)}\right) = 0. \quad (3)$$

The above equation is transcendental and, in general, needs to be solved numerically. We observe that this equation has a single solution if $I_c L_g < \Phi_0/(2\pi)$, which is well satisfied in our circuit.

Consequently, the approximated SQUID loop Lagrangian in terms of the shifted fields reads

$$\mathcal{L}_S' = -\left(\frac{1}{2L_J} + \frac{1}{2L_g}\right)\tilde{\varphi}_g^2 - \frac{\tilde{\varphi}_s^2}{L_J} + g_0^2\tilde{\varphi}_g\tilde{\varphi}_s^2, \quad (4)$$

where we define the optomechanical coupling constant and the Josephson inductance, respectively, as

$$g_0^2 = \frac{4 I_c \pi^2}{\Phi_0^2} \sin\left( \frac{\pi}{\Phi_0} \varphi_g^{(0)} \right), \tag{5}$$

$$L_J^{-1} = \frac{4 I_c \pi}{\Phi_0} \cos\left( \frac{\pi}{\Phi_0} \varphi_g^{(0)} \right). \tag{6}$$

The details of this calculation and the following series expansion, along with the appropriate approximations, are carried out in Appendix A. This approximate Lagrangian $\mathcal{L}_S'$ replaces $\mathcal{L}_S$ in the full Lagrangian which hence becomes $\mathcal{L}' = \mathcal{L}_R + \mathcal{L}_S'$. Note that the resonator Lagrangian (1) retains its original form even though the field variables are replaced by the shifted fields.

The equations of motion corresponding to $\mathcal{L}'$ are obtained by direct application of the Euler–Lagrange equation, which results in

$$\ddot{\tilde{\varphi}}_a + \omega_a^2 (\tilde{\varphi}_a - \tilde{\varphi}_s) + \sum_{f \in [h,c]} \alpha_{fa} \ddot{\tilde{\varphi}}_f = 0, \tag{7a}$$

$$\frac{\tilde{\varphi}_a - \tilde{\varphi}_s}{L_a} - \frac{2\tilde{\varphi}_s}{L_J} + 2 g_0^2 \tilde{\varphi}_s \tilde{\varphi}_g = 0, \tag{7b}$$

$$\ddot{\tilde{\varphi}}_b + \frac{\tilde{\varphi}_b - \tilde{\varphi}_g}{C_b L_b} = 0, \tag{7c}$$

$$\left( \frac{1}{L_b} + \frac{1}{L_g} + \frac{1}{L_J} \right) \tilde{\varphi}_g - \frac{\tilde{\varphi}_b}{L_b} - g_0^2 \tilde{\varphi}_s^2 = 0, \tag{7d}$$

$$\ddot{\varphi}_f + \omega_f^2 \varphi_f + 2 \gamma_f \dot{\varphi}_f + \alpha_f \ddot{\tilde{\varphi}}_a = \xi_f(t), \tag{7e}$$

where $f = h, c$ for hot and cold filter resonator, respectively, we define the angular frequencies $\omega_a = 1/\sqrt{C_{\Sigma a} L_a}$ and $\omega_f = 1/\sqrt{C_{\Sigma f} L_f}$, and the dimensionless coupling constants $\alpha_{fa} = C_{fa}/C_{\Sigma a}$ and $\alpha_f = C_{fa}/C_{\Sigma f}$, the filter dissipation rate $\gamma_f = 1/(2 R_f C_{\Sigma f})$, along with the noise source function $\xi_f(t)$ due to the thermal environment [62], characterized by the power spectral density [63]:

$$S_f(\omega) = \int_{-\infty}^{\infty} \left\langle \xi_f(t+\tau) \xi_f(t) \right\rangle e^{i\omega\tau} \, d\tau = 4 \hbar \omega \gamma_f^2 R_f \coth\left( \frac{\hbar \omega}{2 k_B T_f} \right), \tag{8}$$

where $T_f$ is the temperature of the heat reservoir, and $k_B$ is the Boltzmann constant. Above, we formally define $\gamma_f$ in terms of the resistance $R_f$, serving as the source of thermal noise in the filter resonator. Since the resistance can be chosen freely, $\gamma_f$ effectively becomes a free parameter. Moving forward, we will treat $\gamma_f$ as a fundamental parameter and infer the corresponding resistance when necessary.

As stated above, our primary focus lies in the evolution of the field $\tilde{\varphi}_b$. To this end, we aim to reduce the above set of equations by integrating out the other degrees of freedom. First, we solve $\tilde{\varphi}_g$ from Eq. (7d) in the time domain and insert the solution into Eqs. (7b) and (7c). In accordance with the approximations in Appendix A, we drop the resulting third-order terms. Next, we solve $\tilde{\varphi}_f$ from Eq. (7e) via Fourier transformation and insert the solution into the Fourier transformed Eq. (7a). The resulting equation for $\tilde{\varphi}_a$ in Fourier space is then solved, and the solution is substituted back into Eq. (7b), thus reducing the number of equations to two. To proceed, we redefine the units in the two remaining equations to render the field variables dimensionless. We obtain the relation $\tilde{\varphi}_{s/b} = (\Phi_0/\pi) \phi_{s/b}$. The details of the above derivation can be found in Appendix B. After these steps, we express the equations of motion

governing the time evolution of the fields $\phi_b$ and $\phi_s$ as

$$\omega_s^2 \phi_s(t) - 2g_s^2 \phi_s(t)\phi_b(t) - \int_{-\infty}^{\infty} \mathcal{K}(t-\tau)\phi_s(\tau)\,\mathrm{d}\tau = \xi(t), \tag{9}$$

$$\ddot{\phi}_b(t) + \omega_b^2 \phi_b(t) - g_b^2 \phi_s^2(t) = 0, \tag{10}$$

where the angular frequencies are defined as $\omega_s^2 = (1 + 2L_a/L_J)\,\omega_a^2$ and $\omega_b^2 = (1 - N_L)/(L_b C_b)$, with $N_L = \left(1 + L_b/L_g + L_b/L_J\right)^{-1}$, and the non-linear coupling constants are given by

$$g_s^2 = \frac{\Phi_0 N_L g_0^2}{\pi C_{\Sigma a}}, \qquad g_b^2 = \frac{\Phi_0 N_L g_0^2}{\pi C_b}, \tag{11}$$

the memory kernel $\mathcal{K}(\omega)$ is defined, in the frequency domain, as

$$\mathcal{K}(\omega) = \frac{\omega_a^4}{\omega_a^2 - \omega^2 \left[1 + \sum_f \alpha_f \wp_f(\omega)\right]}, \tag{12}$$

where

$$\wp_f(\omega) = \frac{\alpha_{fa}\omega^2}{\omega_f^2 - \omega^2 - 2\mathrm{i}\gamma_f \omega}, \tag{13}$$

and the noise power spectral density is given by

$$S(\omega) = \frac{\omega_a^4 \sum_f |\wp_f(\omega)|^2 S_f(\omega)}{\left|\omega_a^2 - \omega^2 \left[1 + \sum_f \alpha_f \wp_f(\omega)\right]\right|^2}. \tag{14}$$

See Appendix B for the details of obtaining these definitions.

Above, we have reduced the original set of equations of motion into two equations by integrating out the noise-driven filter modes along with $\varphi_a$, and consolidating the two modes within resonators B. We are left with Eq. (9), describing the dynamics at the node coupling the SQUID to resonator A, and Eq. (10), governing the dynamics of resonator B. Note that Eq.(10) retains the form of a harmonic oscillator equation, while Eq. (9) is governed by the memory kernel $\mathcal{K}(\omega)$ and the noise source $\xi(\omega)$. Let us define the effective frequency of the flux-tunable resonator consisting of the bare resonator A and the SQUID termination, ignoring resonator B and coupling to the filters, as depicted in Fig. 2. This is the normal mode frequency of Eq. (9), derived by assuming vanishing coupling to the filters, $\alpha_f = 0$, and a constant field $\phi_b$ in resonator B, given by

$$\omega_a'(\phi_b) = \frac{1}{\sqrt{C_{\Sigma a}\left[L_a + L_J\left(2 - 2g_s^2 L_J C_{\Sigma a}\phi_b\right)^{-1}\right]}}. \tag{15}$$

This is the angular frequency to be tuned into the operational range, given by $\omega_c < \omega_a' < \omega_h$, as discussed in Sec. 2.

### 3.1.1 Time-independent Green's function

Our next step is to integrate out the remaining high-frequency components in the system by solving Eq. (9) for $\phi_s$. Before attempting a solution to Eq. (9), let us try to gain some intuition into the equation and the implied dynamics by examining its structure. By assuming the field $\phi_b$ is a time-independent constant, we recover the time-independent Green's function for Eq. (9) in the frequency domain as

$$G_0(\omega) = \left[\omega_s^2 - 2g_s^2 \phi_b - \mathcal{K}(\omega)\right]^{-1}. \tag{16}$$

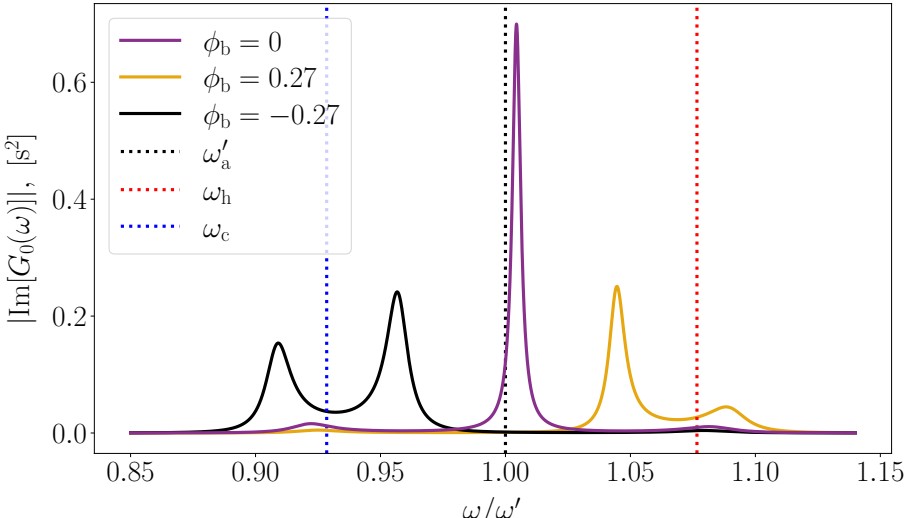

Figure 3: Absolute value of the imaginary part of the time-independent Green's function $G_0(\omega)$ of the SQUID mode as a function of angular frequency at different time-independent values of $\phi_b$. The red and blue vertical dashed lines represent the bare frequencies of hot and cold filters, respectively, while the black dashed line shows the flux tunable angular frequency $\omega_a'$, defined in the main text. We demonstrate the dependence of the Green's function on $\phi_b$ by using the values $\phi_b \in \{0, \pm 0.27\}$. The parameters used for these results are listed in Table 1.

In Fig. 3, we show the absolute value of the imaginary part of the above Green's function at different values of $\phi_b$. We observe the peaks related to each of the modes $\phi_s$, $\phi_h$, and $\phi_c$, slightly offset from their estimated bare values due to their coupling energies. As expected, the peak associated with $\phi_s$ is clearly the strongest. The mode $\phi_s$ serves as an auxiliary mode, housed by resonator A, to model the SQUID-mediated coupling between resonators A and B. The coupling to the filters is mediated by the mode $\phi_a$, and further weakened by the relatively weak capacitive coupling. We demonstrate the effect of the field $\phi_b$ on Eq. (9) by showing the Green's function with $\phi_b = 0$ and $\pm 0.27$. As alluded to in Sec. 2, the phenomenological notion of $\phi_b$ modulating the angular frequency of resonator A between the filter frequencies is clearly evinced here. Depending on the value of $\phi_b$, the peak related to $\phi_s$ moves towards the corresponding filter peak. We also observe a slight repulsion from the filter peak, causing it to shift further away from the $\phi_s$ peak as it approaches.

## 3.2 Finding the time-dependent Green's function

Equation (9) is a time-non-local integral equation with a peaked-spectrum noise function as the source term. We therefore employ Green's function methods to find the solution for an arbitrary source. We treat the left side of the equation (9) as a differential operator, so that the Green's function obeys the following equation:

$$\left[\omega_s^2 - 2g_s^2 \phi_b(t)\right] G(t, t') - \int_{-\infty}^{\infty} \mathcal{K}(t - \tau) G(\tau, t') \, d\tau = \delta(t - t'). \tag{17}$$

The above equation is challenging to solve due to the time dependence of $\phi_b(t)$, and its closed-form solution remains unavailable in the general case. In order to proceed, we express the field variable $\phi_b$ as

$$\phi_b(t) = 2A_b(t) \cos[\omega_b t + \theta_b(t)], \tag{18}$$

where $A_b(t)$ and $\theta_b(t)$ are the time-dependent amplitude and phase of the field, respectively. In the following computation, we invoke the approximation of the slow temporal evolution of $A_b(t)$ and $\theta_b(t)$. This is formally expressed as the assumption that the evolution rates are much lower than the inverse of the heat bath correlation time: $\dot{A}_b,\ \dot{\theta}_b \ll \gamma_f, \omega_b$. In practice, however, this simply means that we ignore the time dependence of $A_b(t)$ and $\theta_b(t)$ in solving the Green's function. To proceed, we transform the equation into the Fourier space by writing the Green's function in a doubly Fourier-transformed form,

$$G(t,t') = \frac{1}{2\pi} \iint_{-\infty}^{\infty} G(\omega,\omega') e^{-i\omega t + i\omega' t'} \, d\omega \, d\omega' , \tag{19}$$

and substitute it in Eq. (17) taking care of the time-dependent term of $\phi_b$. We arrive at the following equation

$$\left[\omega_s^2 - \mathcal{K}(\omega)\right] G(\omega,\omega') - 2g_s^2 A_b \left[e^{-i\theta_b} G(\omega-\omega_b,\omega') + e^{i\theta_b} G(\omega+\omega_b,\omega')\right] = \delta(\omega-\omega'). \tag{20}$$

To facilitate numerical solutions, let us reformulate the problem as a matrix equation. Specifically, we look for solutions in the vicinity of the integer multiples of the angular frequency, $\omega_b$, in Fourier space. To this end, we express the Green's function in a form

$$G(\omega,\omega') = \sum_n G_n(\omega)\delta(\omega-\omega'-n\omega_b). \tag{21}$$

By substituting this Ansatz into Eq. (20), one arrives at a set of equations:

$$P(\omega+n\omega_b)G_n(\omega) + R^*(\theta_b)G_{n-1}(\omega) + R(\theta_b)G_{n+1}(\omega) = \delta_{n,0}, \qquad n = 0, \pm 1, \pm 2, \ldots, \tag{22}$$

where $P(\omega) = \omega_s^2 - \mathcal{K}(\omega)$ and $R(\theta_b) = -2g_s^2 A_b e^{i\theta_b}$. This formulation produces a system of equations, with each index $n$ corresponding to one equation. The entire system can be expressed as a matrix equation, which can be solved efficiently numerically (see Appendix C). By solving the matrix equation to a sufficient degree in $n$, a large enough set of coefficients $G_n(\omega)$ in the series representation of the Green's function can be determined, allowing accurate evaluation of the Green's function. In this work, we set the range $n \in [-1000, 1000]$ for all computations, which is more than enough for reaching convergence. Once the Green's function is known, one can solve the equation (9) for an arbitrary source $\xi(\tau)$:

$$\phi_s(t) = \int_{-\infty}^{\infty} G(t',t)\xi(t') \, dt' . \tag{23}$$

### 3.3  Averaging over noise and time

Considering equation (10), we note that it actually does not depend linearly on $\phi_s$, but rather its square. Furthermore, as mentioned above, the characteristic frequency of the oscillations of $\phi_{s/a}$ is far-off-resonant, and at a much higher frequency, as compared to the oscillations of $\phi_b$. In addition, despite $\phi_s$ being a pure noise with zero average, its square has a non-vanishing expectation value. The typical frequencies of the fluctuations around this mean value are given by the frequency of working mode A which is highly off-resonant to mode B. Therefore, we replace the $\phi_s^2$ appearing in Eq. (10) by its expectation value over noise realizations. Within this approximation, we neglect possible heat transfer between modes A and B and the energy exchange between the occurs only through performing work on or by mode A. The resulting noise-averaged equation for the field $\phi_b$ reads as

$$\ddot{\phi}_b + \omega_b^2 \phi_b + 2\gamma_b \dot{\phi}_b - g_b^2 \left\langle \phi_s^2 \right\rangle_\xi (t) = 0 , \tag{24}$$

where we have ad hoc introduced dissipation into the equation, determined by the decay rate $\gamma_b$. This allows studying the effects of varying loss rates, including the effect of power output when the work done by the QHE is utilized. Here, we focus only on coherent dynamics of the slow mode and neglect the respective noise term related to the introduced dissipation.

Next, we use the expression (18) once more by inserting it into the above equation, and time averaging over one period of the mode $\phi_b$. Here, we consider a single Fourier harmonic of $\langle \phi_s^2 \rangle_\xi (t)$ resonant to the mode $\phi_b$, neglecting contributions from the off-resonant Fourier harmonics. We notice that $\langle \phi_s^2 \rangle_\xi (t)$ has a non-vanishing zero-frequency Fourier component due to its positivity. This component results in a constant shift to the solution of Eq. (24) which cannot be accounted for by Ansatz (18) without violating the assumption of slow dynamics of amplitude $A_b(t)$ and phase $\theta_b(t)$. Physically, this shift corresponds to correction to a direct current flowing through the SQUID and, thus, shift of the equilibrium point for mode B away from the classical potential minimum (3). In practice, this correction turns out to be small and can be neglected. At this point, we invoke the final assumption regarding the time scales of the system: the field amplitude $A_b(t)$ evolves slowly in time compared with the oscillation frequency $\omega_b$. To simplify the equation and to focus on the leading-order behaviour, we neglect all second-order derivatives, products of derivatives, and other small terms in the spirit of the WKB approximation [61]. Subsequently, we decompose the result into its real and imaginary components, yielding separate coupled equations of motion for the amplitude and phase, given as

$$\dot{A}_b(t) + \gamma_b A_b(t) + \frac{g_b^2}{2\omega_b} \mathrm{Im}\left[\langle \phi_s^2 \rangle_{\xi,t} (A_b, \theta_b)\right] = 0 \,, \tag{25}$$

$$A_b(t) \dot{\theta}_b(t) + \frac{g_b^2}{2\omega_b} \mathrm{Re}\left[\langle \phi_s^2 \rangle_{\xi,t} (A_b, \theta_b)\right] = 0 \,, \tag{26}$$

where

$$\langle \phi_s^2 \rangle_{\xi,t} (A_b, \theta_b) = \frac{1}{2\pi} e^{i\theta_b} \int_{-\infty}^{\infty} \sum_n G_n(\omega) G_{n-1}^*(\omega) S(\omega) \, d\omega \,, \tag{27}$$

is given in terms of the Green's function coefficients $G_n(\omega)$, the dependence of which on $A_b$ and $\theta_b$ is visible in Eq. (22). The details of this calculation, along with the associated approximations, are provided in Appendix D. This expression can be evaluated efficiently numerically by the method described above. The equations of motion, Eqs. (25) and (26), yield the dynamics of resonator B at the slowest time scale of the circuit. We proceed with the analysis of their solution in the next section.

# 4 Numerical demonstration of an autonomous quantum heat engine

## 4.1 Microwave power generation rate

As explained in the beginning, our primary focus is to determine whether the proposed circuit can induce coherent generation of microwaves in resonator B and to identify the conditions under which this occurs. Although Eq. (25) needs to be solved numerically in general, its simple structure allows us to obtain certain results without explicitly solving it. Considering Eq. (25) as a generalized Langevin equation [64], we can define the amplitude-dependent total dissipation rate as

$$\Gamma_{\mathrm{tot}}(A_b, \theta_b) = \gamma_b + \frac{g_b^2}{2A_b \omega_b} \mathrm{Im}\left[\langle \phi_s^2 \rangle_{\xi,t} (A_b, \theta_b)\right]. \tag{28}$$

Table 1: Physical parameters of the circuit used in the calculations herein. In the left section, we provide the values of the elementary parameters of the circuit labelled in Fig. 2, and the right section gives the derivative parameters defined throughout the main text.

| Elementary parameters | | | | Derivative parameters | | | |
|---|---|---|---|---|---|---|---|
| $L_a$ | 0.55 nH | $C_a$ | 0.18 pF | $\omega_a/(2\pi)$ | 15 GHz | $\omega'_a/(2\pi)\|_{\phi_b=0}$ | 10.03 GHz |
| $L_h$ | 0.75 nH | $C_h$ | 0.27 pF | $\omega_h/(2\pi)$ | 11.0 GHz | $\omega_s/(2\pi)$ | 20.2 GHz |
| $L_c$ | 1.03 nH | $C_c$ | 0.37 pF | $\omega_c/(2\pi)$ | 8.03 GHz | $L_J$ | 1.36 nH |
| $L_b$ | 0.78 nH | $C_b$ | 0.2 nF | $\omega_b/(2\pi)$ | 379 MHz | $N_L$ | 0.103 |
| $L_g$ | 96.5 pH | $C_{ha}$ | 9.5 fF | $\alpha_h$ | 0.034 | $\alpha_{ha}$ | 0.046 |
| $I_c$ | 0.8 μA | $C_{ca}$ | 16.1 fF | $\alpha_c$ | 0.042 | $\alpha_{ca}$ | 0.078 |
| $T_c$ | 10 mK | $\gamma_{h/c}$ | $\omega_{h/c}/85$ | $g_0$ | 44.5 $\sqrt{A}$/Wb | $g_b$ | $0.66 \times \omega_b$ |
| $T_h$ | 300 mK | $\Phi_{ext}$ | $0.5253 \times \Phi_0$ | $\varphi_g^{(0)}$ | $0.45 \times \Phi_0$ | $g_s$ | $0.52 \times \omega_a$ |

From this, we observe that when the total dissipation rate is positive, the amplitude decays in time. In contrast, a negative total dissipation rate causes the amplitude to grow. This is the condition for coherent microwave generation. Here, we refer to $\gamma_b$ as the intrinsic dissipation rate since it contains all the sources of dissipation, be it internal or external, apart from the effect of the average noise pressure induced by the coupling to resonator A. In addition, we define the intrinsic quality factor as $Q_b = \omega_b/\gamma_b$.

First and foremost, we verify that there exists a range of parameters that enables a negative total dissipation rate. In Fig. 4(a), we show the total dissipation rate, given by Eq. (28), as a function of $A_b$ for various intrinsic quality factors $Q_b$ and other parameters given in Table 1. Notably, the total dissipation rate can become locally negative within a certain range of $A_b$, depending on the intrinsic dissipation rate. Moreover, the total dissipation rate reaches large enough negative values to facilitate its feasible experimental observation. For example, the curve corresponding to an intrinsic quality factor of $Q_b = 13600$, which is not a very high internal quality factor by modern standards [65–69], still reaches negative values. Note that the parameters of Table 1 used here are not optimized for maximum negative dissipation rates, but roughly chosen to provide maximum power output, as explained below. The choice of parameters was based on experimental feasibility and maximising the chances of successful experimental observation upon a physical realisation. It is possible to obtain even higher negative total dissipation rates at the cost of lowering the output power. Nevertheless, the parameters of Table 1 are applied to all subsequent computations unless otherwise explicitly stated.

To study the dynamics further, we use Eq. (25) to analyse how the field amplitude $A_b$ evolves as a function of itself. Although there are multiple minima, referred to as negative-dissipation valleys, present in the curves in Fig. 4(a), we focus on the deepest, most prominent valley, occurring at the lowest amplitudes. We refer to it as the primary dissipation valley. Whenever the total dissipation rate is positive, the time derivative of the field amplitude $A_b$ is negative, resulting in a decreasing amplitude in time. Conversely, when the total dissipation rate becomes negative, the amplitude increases with time. Since the total dissipation rate can be negative only within a finite negative dissipation valley, the amplitude increases until it reaches the point where $\Gamma_{tot} = 0$. Consequently, we can identify stable points along the curves at the zero crossings, where the slope of the curve is positive. In addition, unstable stationary points exist at the zero crossings, where the slope is negative. Near these points, the amplitude will either decay to zero or start to increase towards the following stable point. In Fig. 4(a), both the stable and unstable stationary points are marked in the curves that have



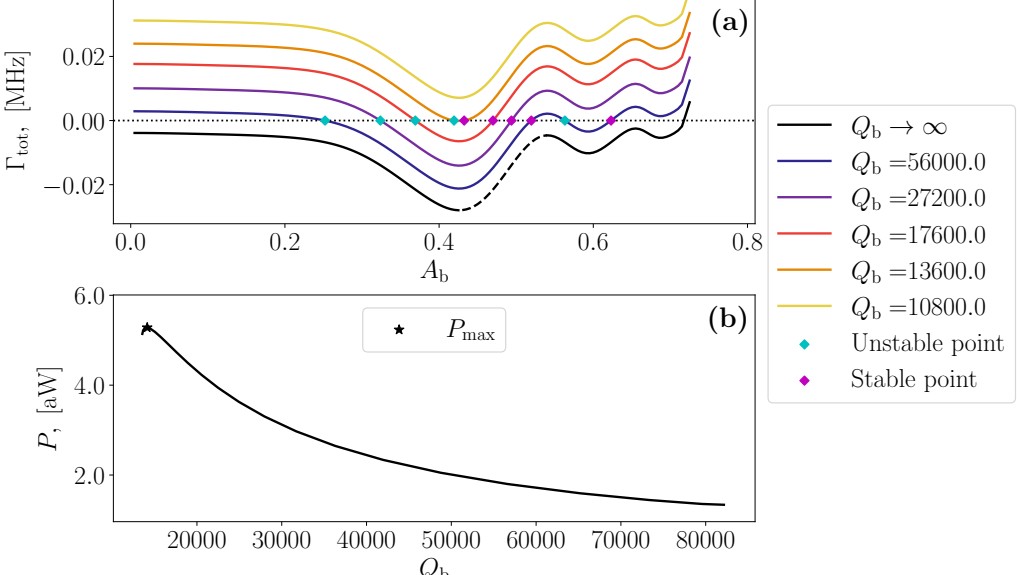

Figure 4: (a) Total dissipation rate $\Gamma_{\text{tot}}(A_{\text{b}}, \theta_{\text{b}})$ as a function of field amplitude at different intrinsic quality factors $Q_{\text{b}}$. The stable and unstable stationary points are marked by the cyan and magenta diamonds, respectively. The dashed line segment on the $Q_{\text{b}} \to \infty$ curve denotes the region where stable points essentially occur in the primary dissipation valley for some $Q_{\text{b}}$. (b) Output power as a function of the intrinsic quality factor, corresponding to the dashed line segment in panel (a). The star marks the location of the maximum value of output power. The parameters used for computing the results are found in Table 1.

zero crossings. This is precisely what we aimed to demonstrate: the average noise pressure causing coherent microwave generation arising solely from the internal dynamics of the system driven by thermal noise.

From Fig. 4(a), we also observe that unless the total quality factor of resonator B, $Q_{\text{b}}$, is very high, there is positive total dissipation at low $A_{\text{b}}$. Therefore, in a practical scenario where decent dissipation from resonator B is allowed, the device has an initiation threshold that needs to be overcome to reach the stable operation point with self-sustained oscillations in resonator B. In this case, only the initial amplitude needs to be prepared by an external agent. In the case of a high quality factor $Q_{\text{b}}$, it is possible that the thermal fluctuations are sufficient to initiate the device, since the dissipation is negative even if $A_{\text{b}}$ approaches zero. However, that implies a very weak coupling of the resonator B to its output line, which significantly lowers the resulting output power in the operating steady state.

As a final remark, we point out that the assumption made above about the rate of change of the field amplitude in resonator B being slow as compared to the other characteristic rates of change holds well. Expressing this condition as $|\Gamma_{\text{tot}}| \sim \dot{A}_{\text{b}}, \dot{\theta}_{\text{b}} \ll \gamma_f, \omega_{\text{b}}$, we find from Fig. 4(a), that even at the largest values of negative dissipation this assumption is fulfilled by orders of magnitude.

### 4.1.1 Power and efficiency

In order to estimate the output power of the device, let us assume that the internal losses of resonator B are negligible. This is reasonable, since, as mentioned above, the intrinsic quality factors considered here are much lower than the internal quality factors of the state-of-the-art resonators in circuit quantum electrodynamics (cQED) [65–69]. Consequently, resonator

B loses energy at a relative rate $\gamma_b$ to an external channel, for instance, a transmission line coupled to the resonator. Utilizing this out-flowing energy as the power output of the engine, we define the output power as

$$P = \gamma_b E_b = 2\gamma_b A_b^2 \frac{(1-N_L)\Phi_0^2}{\pi^2 L_b}, \tag{29}$$

where $E_b$ is the energy stored in resonator B. Based on this, we can compute the attainable output power for all possible stable points by finding the pairs of intrinsic quality factors $Q_b$ and field amplitudes $A_b$ that satisfy the condition $\Gamma_{tot} = 0$ and $\partial_{A_b}\Gamma_{tot} > 0$. This is demonstrated in Figs. 4(a),(b), where panel (b) shows the power as a function of $Q_b$, determined in the range of possible stable points in the primary dissipation valley, depicted in panel (a). Note that because of the $A_b^2$-dependence of power, the parameters yielding maximal power output may slightly differ from those yielding the deepest primary dissipation valley.

The efficiency of a heat engine is typically defined as the ratio of the work output per cycle to the heat absorbed from the hot heat bath per cycle. This definition assumes that heat transfer occurs solely through the working body, meaning that the hot and cold environments are completely decoupled except for their interactions with the working body. However, this assumption does not hold for the present device. Although weak compared to other interactions in the system, there is a continuous coupling between the filter resonators, even though they are off-resonant. Therefore, in order to obtain a realistic estimate for the efficiency, we must account for the continuous heat flow through the system and alter the definition of efficiency accordingly. Consequently, we define efficiency as

$$\eta = \frac{P}{\left|\langle \dot{Q} \rangle\right|}, \tag{30}$$

where $P$ is the power computed above, and $\langle \dot{Q} \rangle$ is the average heat flow through the device. Since the non-linear interaction with resonator B is assumed to be a weak perturbation to the linear part of the system, we assume that the heat flow remains largely unaffected by the modulation due to resonator B. Therefore, we estimate the average heat flow based on a linear system, neglecting resonator B and treating the combined system of Resonator A and the SQUID loop as a single resonator with the angular frequency $\omega_a'\big|_{\phi_b=0} = 1/\sqrt{C_{\Sigma a}(L_a + L_J/2)}$. The average heat flow is given by

$$\langle \dot{Q} \rangle = \sum_{f \in [h,c]} \frac{\alpha_{fa}}{\alpha_f} \left( \langle \xi_f(t)\dot{\phi}_f \rangle - 2\gamma_f \langle \dot{\phi}_f^2 \rangle \right), \tag{31}$$

which can be evaluated using similar methods as above. This result and the method for evaluating it are presented in Appendix E.

We find the efficiency of our QHE to reside in the below 1% range, as shown below. This means that over 99% of the heat just flows straight through the device into the cold reservoir. We additionally test the effect of varying the angular frequency $\omega_a'$, and find that it barely affects the estimated efficiency. These observations validate our method of estimating the efficiency. For completeness, we denote the value of $\eta_C = 1 - T_c/T_h \approx 97\%$ for the Carnot efficiency based on the temperatures listed in Table 1. Additionally, depending on the field amplitude $A_b$, we find the Otto efficiency to reside in the range $\eta_O = 1 - \omega_{a,min}'/\omega_{a,max}' \in [10\%, 50\%]$, where $\omega_{a,min}'$ and $\omega_{a,max}'$ are the minimum and maximum values of the effective working body angular frequency over the cycle, respectively. (See Sec. 4.4.) We find that the efficiency of our device is significantly lower than either of these theoretical bounds. In principle, the efficiency can be improved by fine-tuning the parameters of the circuit, such as coupling between the filters and resonator A to facilitate heat exchange, filter frequency separation, and frequency of the slow resonator. In this work, however, we merely demonstrate a proof-of-principle possibility of autonomous QHE operation with experimentally feasibly observable output power.

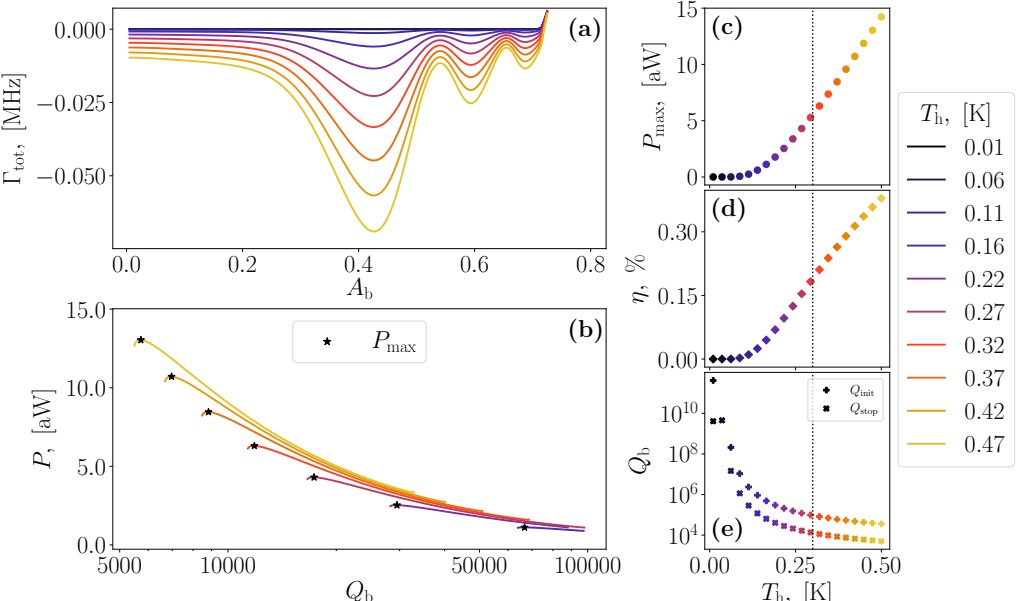

Figure 5: (a) Total dissipation rate $\Gamma_{\text{tot}}(A_b, \theta_b)$ as a function of the field amplitude $A_b$ at various indicated hot-bath temperatures $T_h$ and for an intrinsic dissipation rate of $\gamma_b = 0$. (b) Output power as a function of $Q_b$, determined from the stable points of the primary dissipation valley. The black stars mark the maximum values for each temperature. Note that the three lowest-temperature curves are cut out because we limit $Q_b$ to 100000. The data is presented on a logarithmic horizontal axis. (c) Maximum attainable output power as a function of $T_h$. (d) Efficiency of the heat engine corresponding to the data in panel (c). (e) Lowest intrinsic quality factor of resonator B required for the heat engine to start spontaneously ($Q_{\text{init}}$ marked by the plus signs) and the lowest possible $Q_b$ enabling self-sustained oscillations ($Q_{\text{stop}}$ marked by the crosses) as functions of temperature. The dotted vertical lines in panels (c), (d), and (e) indicate the value of $T_h$ given in Table 1. In panels (a) and (b), we only show every other curve, as compared with the markers in panels (c) and (d), for the sake of clarity. The values in the legend are in units of kelvin, and the parameters used for computing these results, not given here, are found in Table 1.

## 4.2 Parameter sweeps

In addition to showing that the device can reach a stable operation point where coherent power generation occurs, it is insightful to analyse the dependence of the output power on the parameters of the circuit. Let us study this by carrying out parameter sweeps, keeping all other parameters at fixed values, given in Table 1, while varying one parameter at a time.

The temperature of the hot bath is, perhaps, the most fundamental parameter to vary. In Fig. 5, we present the total dissipation, power, and efficiency at different hot-bath temperatures. In panel (a), we show the total dissipation rate with $\gamma_b = 0$ as a function of the field amplitude at different hot reservoir temperatures while keeping the cold reservoir at $T_c = 10$ mK. We observe that the primary dissipation valley deforms as a function of the hot-bath temperature. Specifically, it deepens with rising temperature $T_h$. Consequently, the maximum power output of the device increases monotonically with increasing temperature, as visible in panel (b), where we show the output power as a function of $Q_b$. In panel (c), we display the maxima of the output power as functions of $T_h$. These data further show that the power of the device increases monotonically with increasing temperature. We also observe that as the

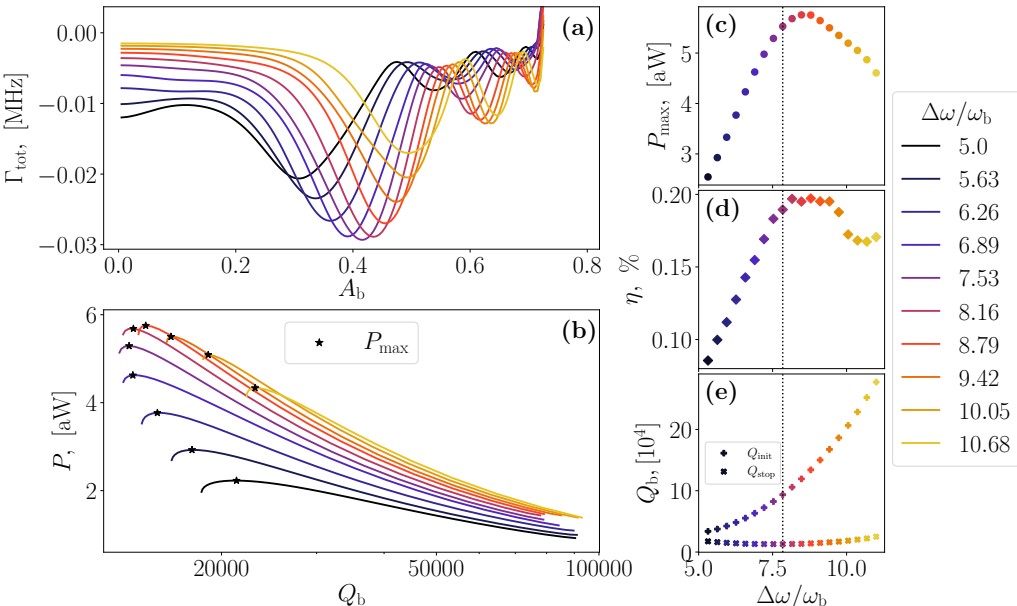

Figure 6: (a) Total dissipation rate $\Gamma_{\text{tot}}(A_{\text{b}}, \theta_{\text{b}})$ as a function of field amplitude $A_{\text{b}}$ at various filter resonator frequency differences $\Delta\omega = \omega_{\text{h}} - \omega_{\text{c}}$ as indicated and for an intrinsic dissipation rate of $\gamma_{\text{b}} = 0$. (b) Output power as a function of $Q_{\text{b}}$, determined from the possible stable points of the primary dissipation valley. The data is presented on a logarithmic horizontal axis. (c) Maximum attainable output power as a function of $\Delta\omega$. (d) Efficiency of the heat engine corresponding to the data in panel (c). (e) Lowest intrinsic quality factor of resonator B required for the heat engine to start spontaneously ($Q_{\text{init}}$ marked by the plus signs) and the lowest possible $Q_{\text{b}}$ enabling self-sustained oscillations ($Q_{\text{stop}}$ marked by the crosses) as functions of $\Delta\omega$. The dotted vertical lines in panels (c), (d), and (e) indicate the difference of filter angular frequencies, given in Table 1. In panels (a) and (b), we show every other curve, as compared to the markers in panels (c) and (d), for the sake of clarity. The parameters used for computing these results that are not given here are found in Table 1.

hot-bath temperature approaches the cold-bath temperature, the work output abates exponentially towards zero, which is an expected consequence of the quantum noise. In panel (d), we display the efficiency as a function of the hot-bath temperature. We note that it follows the trend of the maximum power in panel (c), but starts to become less steep at high temperatures, in agreement with a finite maximum efficiency. In panel (e), we display the lowest intrinsic quality factor of resonator B as a function of $T_{\text{h}}$ required for the spontaneous initiation of power generation in the QHE, as well as the lowest possible $Q_{\text{b}}$ for the device to function as a QHE.

Two parameters that greatly affect the performance of the heat engine are the centre frequencies of the filters, and especially, the angular frequency difference $\Delta\omega = \omega_{\text{h}} - \omega_{\text{c}}$. In Fig. 6, we present total dissipation, power, and efficiency at different angular frequency differences $\Delta\omega$. Panel (a) displays the total dissipation rate with $\gamma_{\text{b}} = 0$ as a function of field amplitude. From here, we observe that increasing the angular frequency difference deforms the primary dissipation valley and moves the position of the minimum higher in $A_{\text{b}}$. We find that there exists an optimal angular frequency difference $\Delta\omega$ that gives the highest negative dissipation rate. This is translated into the attainable output power, as is visible in panel (b), where we show the output power as a function of $Q_{\text{b}}$. It turns out that there exists an optimal

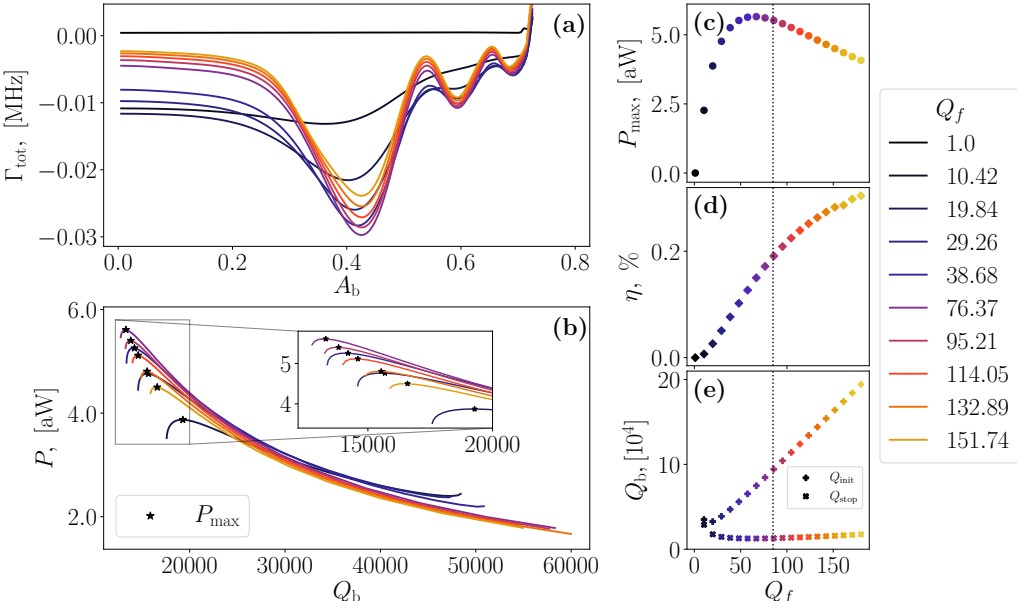

Figure 7: (a) Total dissipation rate $\Gamma_{\text{tot}}(A_{\text{b}}, \theta_{\text{b}})$ as a function of field amplitude $A_{\text{b}}$ at various indicated filter resonator quality factors $Q_f = \omega_f/\gamma_f$ for an intrinsic dissipation rate of $\gamma_{\text{b}} = 0$. We use the quality factor $Q_f$ for both filters. (b) Output power as a function of $Q_{\text{b}}$, determined from the possible stable points of the primary dissipation valley. Note that the curve with $Q_f = 1$ in panel (a) does not produce power output. (c) Maximum attainable output power as a function of $Q_f$. (d) Efficiency of the heat engine corresponding to the data in panel (c). (e) Lowest intrinsic quality factor of resonator B required for the heat engine to start spontaneously ($Q_{\text{init}}$ marked by the plus signs) and the lowest possible $Q_{\text{b}}$ enabling self-sustained oscillations ($Q_{\text{stop}}$ marked by the crosses) as functions of $Q_f$. The dotted vertical lines in panels (c), (d), and (e) mark the value of $Q_f$, given in Table 1. In panels (a) and (b), we only show every other curve, as compared to the markers in panels (c) and (d) for the sake of clarity. The parameters used for computing these results that are not given here are found in Table 1.

value for the angular frequency separation $\Delta\omega$, as evidenced by the data in panel (b). This is even more evident from the data in panel (c), where we display the maximum output power as a function of $\Delta\omega$. One may naively think that increasing $\Delta\omega$ would always lead to an equilibrium with a higher field amplitude $A_{\text{b}}$, thus translating into increasing output power as a function of $\Delta\omega$ [45]. This is not the case here, however. Rather, the optimal value seems to be very close to $\Delta\omega = 8.6 \times \omega_{\text{b}}$, and the output power falls off in both directions from here. The assumption that increasing $\Delta\omega$ increases the field amplitude $A_{\text{b}}$ required to reach a steady state is correct, though, as visible in panel (a). However, the depth of the dissipation valley diminishes with increasing $\Delta\omega$ and hence causes the feasible $\gamma_{\text{b}}$ and the energy generation rate to decrease. Regarding the angular frequency difference, we also note that the chosen value is not optimal, as evinced by the vertical dotted line. In Fig. 6(d), we display the efficiency as a function of the angular frequency difference. We observe that below the optimal value, the efficiency closely follows the trend of the maximum power in panel (c), but starts to deviate at higher $\Delta\omega$. In panel (e), we display the lowest intrinsic quality factor of resonator B as a function of $\Delta\omega$ required for the spontaneous initiation of power generation, as well as the lowest possible $Q_{\text{b}}$ for the device to function as QHE.

To demonstrate the effects of the peaked spectral shape of the filters and the non-Markovian properties of the heat baths, in Fig. 7, we vary the quality factor of the filters, defined as $Q_f = \omega_f/\gamma_f$, which is inversely proportional to the filter linewidth. Here, we use an equal quality factor $Q_f$ for both filters. From panel (a), we find the total dissipation rate $\Gamma_{\text{tot}}$ as a function of the field amplitude $A_b$, while panel (b) shows the attainable output power as a function of $Q_b$. From these data, we clearly see that as the spectrum flattens in the limit $Q_f \to 0$, the negative-dissipation valleys, and therefore the power output, disappear. Although even a modest quality factor $Q_f$ is sufficient to enable power generation, it is evident that the device demands peaked filter spectral shapes in order to function as a quantum heat engine. The peaked spectra effectively provide the mechanism for modulating the coupling of the working body to the heat bath over the cycle, as the angular frequency of the working body is modulated. This is a necessary requirement for realizing an Otto cycle [60]. These conclusions are further illustrated in panel (c), where we plot the maximum output power as a function of $Q_b$. Here, we observe a quick increase in power as the filter spectrum becomes narrower. The data in panel (c) further demonstrates that increasing the filter quality factor too much starts to reduce the output power. This is expected, as a higher $Q_f$ translates to weaker coupling to the thermal noise. In panel (d), we display the efficiency as a function of the filter quality factor $Q_f$. Below the maximum power, the efficiency enhances with increasing $Q_f$, as the output power rises. When the output power starts to diminish, the slope of the efficiency also starts to abate. The above observations confirm that the peaked spectral shape of the filters plays an important role in the dynamics of the device, suggesting that non-Markovian effects should not be neglected. Finally, in panel (e) we show the lowest intrinsic quality factor of resonator B as a function of $Q_f$ required for the spontaneous initiation of generation, as well as the lowest possible $Q_b$ for the device to function as QHE.

## 4.3 Quantum disadvantage

Let us revisit the temperature dependence of the hot heat bath. Above, we alluded that the threshold temperature for microwave generation should be a consequence of the quantum noise. In order to highlight the quantum nature of the device, and to investigate the implications of quantum thermal noise to the system, we repeat the computation done for Fig. 5 using the corresponding power spectral density of classical thermal noise. It is obtained from the approximation valid at the high-temperature limit of the quantum version:

$$S_f^{\text{clas}}(\omega) = 8\gamma_f^2 R_f k_B T_f \approx 4\hbar\omega\gamma_f^2 R_f \coth\left(\frac{\hbar\omega}{2k_B T_f}\right), \quad \text{for} \quad \frac{\hbar\omega}{k_B T_f} \ll 1. \quad (32)$$

This is the only change as compared with how the results in Fig. 5 are obtained.

In Fig. 8, we show the results of the above-defined computation at various hot-bath temperatures. In panel (a), we show the total dissipation rate $\Gamma_{\text{tot}}(A_b, \theta_b)$ as a function of the field amplitude $A_b$ while panel (b) depicts the output power as a function of the intrinsic quality factor $Q_b$. The results in these panels look qualitatively similar to those in Figs. 5(a) and (b), only with higher negative dissipation and output power. In panel (c), however, we show the maximum output power as a function of $T_h$, from where we observe that the initial plateau of exponentially vanishing output power, observed in Fig. 5(c), has disappeared. In the case of classical noise, the maximum power increases linearly with $T_h$ everywhere. This behaviour is different from the quantum case, and can be explained by the linear temperature dependence of the classical noise spectrum, given by Eq. (32). We observe the corresponding qualitative difference in the efficiency, shown in panel (d). The decrease in the slope of efficiency as temperature rises is slightly more prominent here than in the quantum case. To conclude, we stress that the temperature of the hot reservoir is much lower than that required for the classical limit to be accurate, as evinced by the ratio $\hbar\omega_h/(k_B T_h) \approx 1.7$. Therefore, the thermal

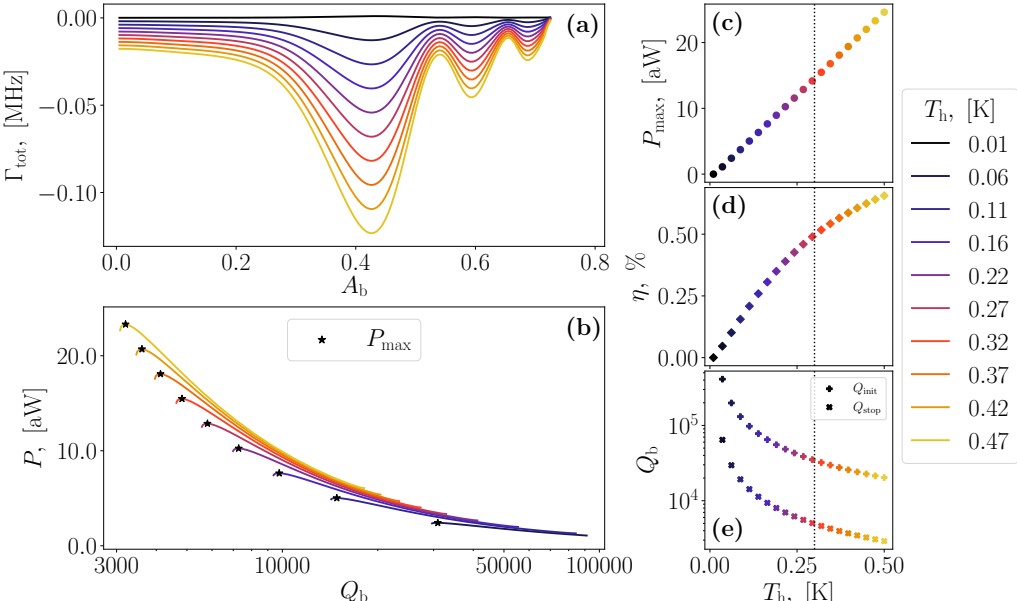

Figure 8: Results in Fig. 5 reproduced using classical noise, characterised by Eq. (32).
(a) Total dissipation rate $\Gamma_{\text{tot}}(A_b, \theta_b)$ as a function of field amplitude $A_b$ at various hot
bath temperatures $T_h$. (b) Output power as a function of intrinsic quality factor as
determined from the possible stable points of the primary dissipation valley. The
data is presented on a logarithmic horizontal axis. (c) Maximum attainable output
power as a function of $T_h$. (d) Efficiency of the heat engine corresponding to the data
in panel (c). (e) Lowest intrinsic quality factor of resonator B required for the heat
engine to start spontaneously ($Q_{\text{init}}$ marked by the plus signs) and the lowest possible
$Q_b$ enabling self-sustained oscillations ($Q_{\text{stop}}$ marked by the crosses) as functions of
temperature. The dotted vertical lines in panels (c), (d), and (e) mark the value of $T_h$
given in Table 1. In panels (a) and (b), we only show every other curve, as compared
to the markers in panels (c) and (d), for the sake of clarity. The values in the legend
are in units of kelvin, and the fixed parameters used for computing these results are
found in Table 1.

baths here need to be described by quantum noise. In panel (e), we show the lowest intrinsic quality factor of resonator B as a function of $T_h$ required for the spontaneous initiation of power generation, as well as the lowest possible $Q_b$ for the device to function as a QHE. Here, similar qualitative behaviour is observed as for the quantum noise.

## 4.4 Comparison to the Otto cycle

Having shown that the device can reach a stable state where power generation is possible, let us analyse the thermodynamic cycle of the device and compare it with the Otto cycle. We study the temporal evolution of the working body by examining its effective photon number. The latter can be found as the ratio of the energy stored in the working body and energy of a single photon, $E_a/(\hbar\omega'_a)$. Since the working body is nighly harmonic, its total energy is given by $E_a = 2\langle E_{a,\text{ind}}\rangle_\xi$, where

$$E_{a,\text{ind}} = \frac{\hbar^2}{e^2} \frac{\phi_s^2}{L_a} \left( \frac{L_a}{L_J} - \frac{g_s^2}{\omega_a^2} \phi_b \right) \left( 1 + \frac{2L_a}{L_J} - 2\frac{g_s^2}{\omega_a^2} \phi_b \right), \tag{33}$$

is the energy of inductive elements comprising resonator A. Here, the noise averaged value for $\phi_s^2$ is provided by Eq. (D.6), and we assume sinusoidal oscillations of $\phi_b$ with amplitude

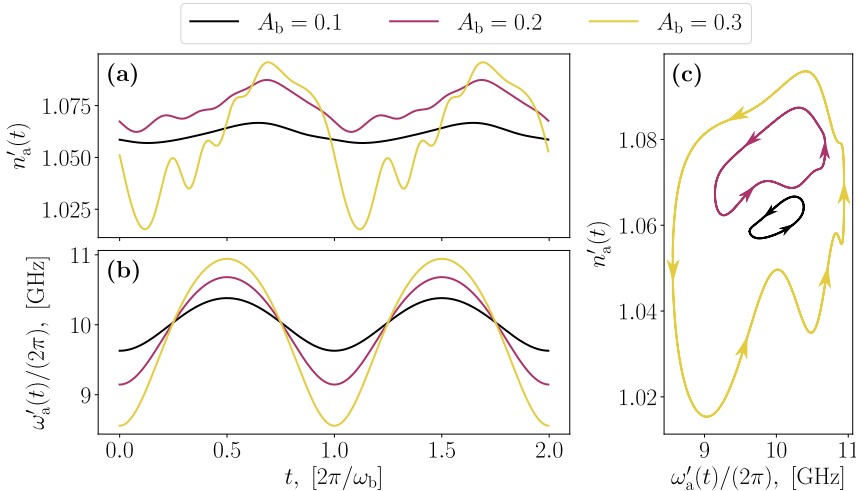

Figure 9: Temporal evolution of the noise expectation value of the (a) effective mean photon number of the working body $n_a(t)$, (b) effective angular frequency of the working body $\omega'_a(t)$, and (c) resulting trajectory of the QHE working body in the $(\omega'_a, n_s)$ plane. The parameters used for computing the results are found in Table 1.

$A_b$. The mean photon number $n'_a(t)$ in the working body, its angular frequency $\omega'_a(t)$, and the trajectory of the working body in the $(\omega'_a, n'_a)$ plane are shown in Fig. 9 for a few field amplitudes $A_b$ in the range where stable points generally occur with the parameters given in Table 1. As a first observation, we notice that the mean photon number shown in Fig. 9(a) exhibits oscillatory behaviour shifted in phase from the oscillations of $\omega'_a$ (see Fig. 9(b)), which matches our expectations and phenomenological understanding [45]. Second, we notice that at high amplitudes the oscillations of the photon number appear peculiar, which may be a signature of underdamped beatings between the working-body mode and the cold filter bath. In Fig. 9(c), we show trajectories in the frequency–photon-number plane, which form closed loops of non-vanishing area. Orientation of these loops is such that mode A produces work, i.e., operates as a heat engine. This strongly corroborates our understanding of the dynamics and the above conclusions.

Although in the externally driven quantum Otto heat cycle, the angular frequency of the working body is ideally square-modulated, it seems reasonable to conclude based on this analysis that the working body of our system, i.e., the effective resonator consisting of resonator A and the SQUID loop, indeed undergoes an approximate cosine-modulated Otto cycle.

## 5 Conclusions

We proposed and theoretically analysed an experimentally feasible superconducting quantum circuit capable of realizing an autonomous quantum heat engine based on internal Otto cycles. To validate the operation of the device, we developed an efficient quasiclassical non-Markovian model to show that the circuit can generate a coherent microwave tone arising purely from heat flow and non-linear internal dynamics. The model enables us to estimate the coherent microwave power generation rate and the efficiency of the device.

We carried out multiple parameter sweeps to demonstrate the effects of the parameters on the dynamics of the device and found the results to be physically sound. Furthermore, we characterized the cyclic evolution of the working body in our device, showing that it realizes a sinusoidally driven Otto cycle. Our findings show that the microwave generation rate can

be relatively high—well within the observable range in circuit quantum electrodynamics—highlighting the feasibility of an experimental realization.

For the experimental observation of the output power, let us consider the ultrasensitive bolometer of Ref. [24] with the noise in its readout signal of $30\,\mathrm{zW}/\sqrt{\mathrm{Hz}}$ in units of the absorbed power and a thermal time constant of $500\,\mathrm{ns}$. These performance metrics yield roughly a $5\,\mu\mathrm{s}$ of integration time to arrive at a signal-to-noise ratio of unity for the $10\,\mathrm{aW}$ output power predicted for the heat engine in Fig. 5, for a hot-bath temperature of only roughly $400\,\mathrm{mK}$. Consequently, channelling the microwaves generated by the quantum heat engine to the bolometer input and observing the bolometer response in time is expected to yield a conveniently fast observation of the generated power directly from the heat it deposits on the bolometer. The coherence of the radiation can be unveiled by a homodyne method at low temperatures before the bolometer [70].

Unlike many theoretical works, we analyse an accurately defined physical device, an electric circuit, and explicitly connect the theoretical model to its circuit parameters. The parameters were chosen such that they ensure experimental feasibility with modern fabrication techniques [59,65,69,71], making this work a solid stepping stone toward autonomous thermal machines in cQED. Given the advancements in thermal devices and quantum thermodynamics within superconducting circuits [30,31,39–43,72], the realization of our proposed quantum heat engine appears to be likely in the near future. For high-power operation, exploring engineered environments [44,73–75], which have recently been shown to enable rapid thermal-state preparation [76], is an interesting direction for future research.

We found the output power of the quantum heat engine to increase monotonically with the temperature of the hot reservoir, providing prospects for potential applications. We also compared the performance of the device when driven by classical noise instead of quantum noise and observed a significant increase in the output power. This observation is attributed to the exponential suppression of thermal excitations in low-temperature heat reservoirs as compared with the linear dependence in the classical case. Our findings may provide insights into the limits and effects of low-temperature thermal environments governed by quantum mechanics, as opposed to classically described thermal baths. In any case, our results pave the way for practical thermal machines in cQED, operating at cryogenic temperatures. For instance, the proposed device may possibly serve as a coherent microwave photon source, harnessing thermal energy from temperature differences within a cryostat. In addition to power, we determine the efficiency of the device and find that it resides in the below 1% range. This is due to the weak, but continuous, coupling between the heat baths, which results in over 99% of the heat simply flowing through the device without contributing to power generation. The efficiency of the device is thus significantly below the Carnot and Otto limits of efficiency. However, the efficiency can potentially be improved by the utilization of high-order filters for the heat baths.

Even though our quasiclassical approach may lead to some missed quantum character, we prioritized retaining the non-linearity and non-Markovianity of the system as the key points of methodological novelty. Fully quantum analysis of such a circuit is technically challenging, although can be done using advanced numerical methods for dissipative superconducting circuits [77]. Quantum effects will undoubtedly introduce corrections to our quasiclassical estimates; however, we do not expect them to eliminate the underlying phenomenon. Investigating the complete quantum nature of the device remains a topic for future work, but the results presented here serve as a significant and compelling proof of concept.

Beyond quantization, another promising theoretical future direction is to extend the analysis of the non-linear coupling to a higher order. On the experimental side, it would be fruitful to explore various realizations of the proposed quantum heat engine and compare their respective advantages and limitations. Finally, we point out that the presented approach does not

directly provide an estimate of work fluctuations; however, it should not be overlooked that phase and amplitude fluctuations in $\phi_b$ are inherently present. Understanding and mitigating these fluctuations in the coherent output field of the heat engine calls for further studies.

## Acknowledgments

We acknowledge the support from the members of the QCD and PICO groups at Aalto University. Especially, we thank J. Pekola, B. Karimi, C. Satrya, M. Tuokkola, T. Mörstedt, J. Ma, H. Kivijärvi, H. Suominen, M. Gunyho, Y. Sunada, Q. Chen, and A. Keränen for fruitful scientific discourse and other help.

**Funding information**   This work was funded by the Academy of Finland Centre of Excellence program (Projects No. 352925 and No. 336810) and Academy of Finland Grants No. 316619 and No. 349594 (THEPOW). We also acknowledge funding from the European Research Council under Advanced Grant No. 101053801 (ConceptQ).

## A   Deriving the optomechanical approximation

To handle the trigonometric potential in the SQUID Lagrangian (2), let us derive an approximation for it. Before expanding the trigonometric functions, however, let us analyse the potential energy related to the inductances in the SQUID loop. The potential energy is given as

$$U = \frac{\left(\varphi_g - \Phi_{ext}\right)^2}{2L_g} - E_J \cos\left(\frac{2\pi \varphi_s}{\Phi_0}\right) - E_J \cos\left(\frac{2\pi}{\Phi_0}\left[\varphi_s - \varphi_g\right]\right), \tag{A.1}$$

where we have chosen to rewrite the external flux, $\Phi_{ext}$, outside of the cosine, as can be done. Let us now find the potential minimum with respect to the fields $\varphi_s$ and $\varphi_g$. By applying $\partial U/\partial \varphi_s = 0$ and $\partial U/\partial \varphi_g = 0$, we find two conditions, the first simply being $\varphi_s^{(0)} = \varphi_g^{(0)}/2$, and the second stating

$$\frac{\varphi_g^{(0)} - \Phi_{ext}}{L_g} + \frac{2E_J \pi}{\Phi_0} \sin\left(\frac{\pi}{\Phi_0}\varphi_g^{(0)}\right) = 0. \tag{A.2}$$

This is a transcendental equation that needs to be solved numerically. This does not matter as we will resort to numerics at the end anyhow.

Let us now assume a solution, $\varphi_g^{(0)}$, of the above equation. We apply a change of variable to the fields inductively coupled to the SQUID in order to shift the fields close to the potential minimum. We define new field variables as

$$\tilde{\varphi}_a = \varphi_a - \varphi_g^{(0)}/2, \qquad \tilde{\varphi}_s = \varphi_s - \varphi_g^{(0)}/2, \tag{A.3}$$

$$\tilde{\varphi}_b = \varphi_b - \varphi_g^{(0)}/2, \qquad \tilde{\varphi}_g = \varphi_g - \varphi_g^{(0)}/2. \tag{A.4}$$

Applying this to the SQUID Lagrangian, one obtains

$$\begin{aligned}
\mathcal{L}_S &= -\frac{\left(\tilde{\varphi}_g + \varphi_g^{(0)} - \Phi_{ext}\right)^2}{2L_g} + E_J \cos\left(\frac{2\pi}{\Phi_0}\left[\tilde{\varphi}_s + \varphi_g^{(0)}/2\right]\right) + E_J \cos\left(\frac{2\pi}{\Phi_0}\left[\tilde{\varphi}_s - \tilde{\varphi}_g - \varphi_g^{(0)}/2\right]\right) \\
&= -\frac{\left(\tilde{\varphi}_g + \varphi_g^{(0)} - \Phi_{ext}\right)^2}{2L_g} + 2E_J \cos\left(\frac{\pi}{\Phi_0}\left[\tilde{\varphi}_g + \varphi_g^{(0)}\right]\right) \cos\left(\frac{\pi}{\Phi_0}\left[2\tilde{\varphi}_s - \tilde{\varphi}_g\right]\right).
\end{aligned} \tag{A.5}$$

Upon expanding the trigonometric functions with respect to $\tilde\varphi_g$ and $\tilde\varphi_s$, we obtain

$$\cos\left(\frac{\pi}{\Phi_0}\left[\tilde\varphi_g + \varphi_g^{(0)}\right]\right)\cos\left(\frac{\pi}{\Phi_0}\left[2\tilde\varphi_s - \tilde\varphi_g\right]\right)$$
$$= \left[\cos\left(\frac{\pi}{\Phi_0}\varphi_g^{(0)}\right) - \frac{\pi}{\Phi_0}\sin\left(\frac{\pi}{\Phi_0}\varphi_g^{(0)}\right)\tilde\varphi_g + \dots\right]$$
$$\times\left\{\left[1 - \frac{2\pi^2}{\Phi_0^2}\tilde\varphi_s^2 + \dots\right]\left[1 - \frac{\pi^2}{2\Phi_0^2}\tilde\varphi_g^2 + \dots\right] + \frac{2\pi^2}{\Phi_0^2}\tilde\varphi_s\tilde\varphi_g + \dots\right\}. \qquad (A.6)$$

Next, we multiply open the parentheses, truncate the expression to second order in field variables everywhere, and plug the result back into the Lagrangian. As we do this, we discover the left-hand side of Eq. (A.2) multiplied by $\tilde\varphi_g$ appearing in the Lagrangian. As this must be zero, we can safely drop it. Further, we will drop all constants, as they will not affect the equations of motion anyhow. The Lagrangian now reads

$$\mathcal{L}_S = 2E_J\left[-\frac{\pi^2}{\Phi_0^2}\cos\left(\frac{\pi}{\Phi_0}\varphi_g^{(0)}\right)\tilde\varphi_g^2 - \frac{2\pi^2}{\Phi_0^2}\cos\left(\frac{\pi}{\Phi_0}\varphi_g^{(0)}\right)\tilde\varphi_s^2 + \frac{2\pi^3}{\Phi_0^3}\sin\left(\frac{\pi}{\Phi_0}\varphi_g^{(0)}\right)\tilde\varphi_g\tilde\varphi_s^2\right.$$
$$\left. + \frac{2\pi^2}{\Phi_0^2}\cos\left(\frac{\pi}{\Phi_0}\varphi_g^{(0)}\right)\tilde\varphi_g\tilde\varphi_s - \frac{2\pi^3}{\Phi_0^3}\sin\left(\frac{\pi}{\Phi_0}\varphi_g^{(0)}\right)\tilde\varphi_g^2\tilde\varphi_s\right] - \frac{\tilde\varphi_g^2}{2L_g}. \qquad (A.7)$$

Above, the first two terms in the brackets give inductive energy terms arising from the SQUID inductance. The third term is the optomechanical coupling, while the fourth and fifth terms are the linear and inverse optomechanical couplings. As a final step, we will drop the linear interaction term as well as the inverse optomechanical coupling, as they are assumed weak due to the large frequency offset between the modes $\tilde\varphi_g$ and $\tilde\varphi_s$. This yields the approximated Lagrangian given by Eq. (4).

## B  Simplification of the equations of motion

As explained in the main text, our goal is to integrate out the field degrees of freedom $\tilde\varphi_a$, $\tilde\varphi_f$ and $\tilde\varphi_g$. We begin by solving $\tilde\varphi_g$ from Eq. (7d) in the time domain as

$$\tilde\varphi_g = N_L\tilde\varphi_b + g_0^2 N_L L_b\tilde\varphi_s^2, \qquad (B.1)$$

where we have introduced $N_L = (1 + L_b/L_g + L_b/L_J)^{-1}$. This solution is then inserted into the Eqs. (7b) and (7c) to obtain

$$\frac{\tilde\varphi_a - \tilde\varphi_s}{L_a} - \frac{2\tilde\varphi_s}{L_J} + 2g_0^2 N_L\tilde\varphi_b\tilde\varphi_s = 0, \qquad (B.2)$$

$$\ddot{\tilde\varphi}_b + \frac{1 - N_L}{C_b L_b}\tilde\varphi_b - \frac{N_L g_0^2}{C_b}\tilde\varphi_s^2 = 0, \qquad (B.3)$$

where we have neglected the third-order term ($\sim \tilde\varphi_s^3$) in Eq. (B.2), in accordance with the truncation of the series discussed in Appendix A. The resulting form of Eq. (B.3), governing the evolution of $\tilde\varphi_b$, readily matches the final shape of Eq. (10) in the main text. Consequently, we only need to combine the three remaining equations.

In order to derive Eq. (9) in the main text, we need to eliminate $\tilde\varphi_f$ and $\tilde\varphi_a$. First, we solve Eq. (7e) for $\tilde\varphi_f$ via Fourier transform, yielding

$$\hat{\tilde\varphi}_f = \frac{\xi_f(\omega) + \alpha_f\omega^2\hat{\tilde\varphi}_a}{\omega_f^2 - \omega^2 - 2i\gamma_f\omega}. \qquad (B.4)$$

We additionally Fourier transform Eq. (7a), and substitute the above solution in there, resulting in

$$-\omega^2 \hat{\tilde{\varphi}}_a + \omega_a^2 \left( \hat{\tilde{\varphi}}_a - \hat{\tilde{\varphi}}_s \right) - \omega^2 \sum_f \alpha_f \wp_f(\omega) \hat{\tilde{\varphi}}_a = \sum_f \wp_f(\omega) \xi_f(\omega), \tag{B.5}$$

where we utilize the helper function $\wp_f(\omega)$, defined in the main text by Eq. (13). The above equation solves to

$$\hat{\tilde{\varphi}}_a = \frac{\omega_a^2 \hat{\tilde{\varphi}}_s + \sum_f \wp_f(\omega) \xi_f(\omega)}{\omega_a^2 - \omega^2 \left[ 1 + \sum_f \alpha_f \wp_f(\omega) \right]}. \tag{B.6}$$

We further Fourier transform Eq. (B.2) and insert the above solution into the obtained equation, finally yielding

$$\left[ \omega_s^2 - \mathcal{K}(\omega) \right] \hat{\tilde{\varphi}}_s(\omega) - \frac{2g_0^2 N_L}{C_{\Sigma a}} \chi(\omega) = \xi(\omega), \tag{B.7}$$

where $\chi(\omega)$ denotes the Fourier transform of $\tilde{\varphi}_s(t)\tilde{\varphi}_b(t)$, and the angular frequency $\omega_s$, the total memory kernel $\mathcal{K}(\omega)$, and the total noise source function $\xi(\omega)$ are defined in the main text. Fourier transforming the above equation back to the time domain yields an equation of the form Eq. (9) in the main text.

As a final step, we transform the field variables to be dimensionless. To this end, we write the field variable in a form $\tilde{\varphi}_i = \sqrt{\hbar Z_i / 2} x_i \phi_i$, where $i = $ s, b, $Z_i$ is the characteristic impedance of the resonator, $\phi_i$ is the dimensionless field variable, and $x_i$ is a dimensional scaling constant. We substitute $\phi_i$ into the equations derived above, and find $x_i$ such that the units match throughout the equation. We find $x_i = \Phi_0/\pi \sqrt{2/(\hbar Z_i)}$. With this, we finally write down the Eqs. (9) and (10) given in the main text.

## C  Numerical solution of the Green's function

To derive Eq. (20) in the main text, we first utilize Eq. (18), where we expand the cosine in terms of exponential functions. Next, we use the Fourier transform of the Green's function, given by Eq. (19) in the main text. Simple substitution of these definitions yields

$$\frac{1}{2\pi} \iint_{-\infty}^{\infty} \left[ \omega_s^2 - \mathcal{K}(\omega) \right] G(\omega, \omega') e^{-i\omega t + i\omega' t'} \, d\omega \, d\omega' \tag{C.1}$$
$$+ \frac{2g_s^2 A_b}{2\pi} \iint_{-\infty}^{\infty} \left( e^{-i[(\omega_b + \omega)t + \theta]} + e^{i[(\omega_b - \omega)t + \theta]} \right) G(\omega, \omega') e^{i\omega' t'} \, d\omega \, d\omega' = \delta(t - t'),$$

where we neglect the time-dependence of $A_b$ and $\theta_b$, as explained in the main text. A change variables ($\omega_b \pm \omega \to \pm\omega$) on the latter row gives

$$\frac{1}{2\pi} \iint_{-\infty}^{\infty} \left[ \omega_s^2 - \mathcal{K}(\omega) \right] G(\omega, \omega') e^{-i\omega t + i\omega' t'} \, d\omega \, d\omega' \tag{C.2}$$
$$+ \frac{2g_s^2 A_b}{2\pi} \iint_{-\infty}^{\infty} \left[ G(\omega - \omega_b, \omega') e^{-i\theta} + G(\omega + \omega_b, \omega') e^{i\theta} \right] e^{-i\omega t + i\omega' t'} \, d\omega \, d\omega' = \delta(t - t').$$

To obtain Eq. (20), we Fourier transform the delta function on the right-hand side and drop the Fourier integrals.

Next, we derive the matrix equation for the Green's function, Eq. (22). Upon inserting the series expansion of the Green's function, given by Eq. (21), into Eq. (20), we obtain

$$\sum_n \left[ P(\omega) G_n + R^*(\theta_b) G_{n-1} + R(\theta_b) G_{n+1} \right] \delta(\omega - \omega' - n\omega_b) = \delta(\omega - \omega'), \tag{C.3}$$

where we shift the indices of the latter two terms by one to write the terms under the same sum. Taking the integral over $\omega$, one finds the Eq. (22) given in the main text. This set of equations can be written as a matrix equation:

$$
\begin{pmatrix}
\cdot & \cdot & \cdot & \cdot & \cdot & \cdot & \cdot \\
\cdot & P(\omega - 2\omega_{\mathrm{b}}) & R(A_{\mathrm{b}}, \theta_{\mathrm{b}}) & 0 & 0 & 0 & \cdot \\
\cdot & R^*(A_{\mathrm{b}}, \theta_{\mathrm{b}}) & P(\omega - \omega_{\mathrm{b}}) & R(A_{\mathrm{b}}, \theta_{\mathrm{b}}) & 0 & 0 & \cdot \\
\cdot & 0 & R^*(A_{\mathrm{b}}, \theta_{\mathrm{b}}) & P(\omega) & R(A_{\mathrm{b}}, \theta_{\mathrm{b}}) & 0 & \cdot \\
\cdot & 0 & 0 & R^*(A_{\mathrm{b}}, \theta_{\mathrm{b}}) & P(\omega + \omega_{\mathrm{b}}) & R(A_{\mathrm{b}}, \theta_{\mathrm{b}}) & \cdot \\
\cdot & 0 & 0 & 0 & R^*(A_{\mathrm{b}}, \theta_{\mathrm{b}}) & P(\omega + 2\omega_{\mathrm{b}}) & \cdot \\
\cdot & \cdot & \cdot & \cdot & \cdot & \cdot & \cdot
\end{pmatrix}
\begin{pmatrix}
\cdot \\ G_{-2} \\ G_{-1} \\ G_0 \\ G_1 \\ G_2 \\ \cdot
\end{pmatrix}
=
\begin{pmatrix}
\cdot \\ 0 \\ 0 \\ 1 \\ 0 \\ 0 \\ \cdot
\end{pmatrix}.
\tag{C.4}
$$

Solving this matrix equation numerically is very efficient, since solving a matrix equation with a banded matrix is a linear time operation in $n$ [78], as opposed to inverting an arbitrary matrix, which has cubic time complexity in $n$.

Once the above matrix equation is solved to the desired degree in $n$, the Green's function can be expressed through Eq. (21) in the frequency domain. This allows us to solve Eq. (9) for an arbitrary source $\xi(t)$. In general, the solution to a differential equation is given as a convolution of the Green's function with the source:

$$
\phi_{\mathrm{s}}(t) = \int_{-\infty}^{\infty} G(t', t) \xi(t') \, \mathrm{d}t'.
\tag{C.5}
$$

# D  Noise and time averages

As explained in the main text, we want to compute the noise expectation value of $\phi_{\mathrm{s}}^2$. Starting from the general solution of $\phi_{\mathrm{s}}$ obtained above, we express this as

$$
\left\langle \phi_{\mathrm{s}}^2 \right\rangle_\xi (t) = \left\langle \iint_{-\infty}^{\infty} G(t', t) \xi(t') G(t'', t) \xi(t'') \, \mathrm{d}t' \, \mathrm{d}t'' \right\rangle.
\tag{D.1}
$$

Let us Fourier transform the noise functions and rearrange the expectation value to obtain

$$
\left\langle \phi_{\mathrm{s}}^2 \right\rangle_\xi (t) = \frac{1}{4\pi^2} \iiiint_{-\infty}^{\infty} G(t', t) G(t'', t) \left\langle \xi(\omega') \xi(\omega'') \right\rangle \mathrm{e}^{-\mathrm{i}\omega' t'} \mathrm{e}^{-\mathrm{i}\omega'' t''} \, \mathrm{d}t' \, \mathrm{d}t'' \, \mathrm{d}\omega' \, \mathrm{d}\omega''.
\tag{D.2}
$$

By the properties of the noise function, we know that the expectation value here is given as $\left\langle \xi_f(\omega) \xi_f(\omega') \right\rangle = 2\pi \delta(\omega + \omega') S_f(\omega)$, where $S_f(\omega)$ is the noise spectral density. Upon inserting this in and taking the $\omega''$ integral, the above becomes

$$
\left\langle \phi_{\mathrm{s}}^2 \right\rangle_\xi (t) = \frac{1}{2\pi} \iiint_{-\infty}^{\infty} G(t', t) G(t'', t) S(\omega') \mathrm{e}^{-\mathrm{i}\omega' t'} \mathrm{e}^{\mathrm{i}\omega' t''} \, \mathrm{d}t' \, \mathrm{d}t'' \, \mathrm{d}\omega',
\tag{D.3}
$$

where we use the property $S(-\omega) = S(\omega)$ of the spectral density. Next, we shall insert the Fourier transform of the Green's function given by Eq. (19), yielding

$$
\left\langle \phi_{\mathrm{s}}^2 \right\rangle_\xi (t) = \left( \frac{1}{2\pi} \right)^3 \int \cdots \int_{-\infty}^{\infty} G(\omega'', \omega) G(\tilde{\omega}, \tilde{\omega}') S(\omega')
$$
$$
\times \mathrm{e}^{-\mathrm{i}\omega'' t' + \mathrm{i}\omega t} \mathrm{e}^{-\mathrm{i}\tilde{\omega} t'' + \mathrm{i}\tilde{\omega}' t} \mathrm{e}^{-\mathrm{i}\omega' t' + \mathrm{i}\omega' t''} \, \mathrm{d}t' \, \mathrm{d}t'' \, \mathrm{d}\omega \, \mathrm{d}\omega' \, \mathrm{d}\omega'' \, \mathrm{d}\tilde{\omega} \, \mathrm{d}\tilde{\omega}'.
\tag{D.4}
$$

We note that both $t'$ and $t''$ integrals can be used to yield Dirac delta functions. We first use the $t'$ integral to produce $\delta(\omega' + \omega'')$ and take the $\omega''$ integral immediately. After this, we utilize the $t''$ integral to produce $\delta(\tilde{\omega} - \omega')$ and take the $\tilde{\omega}$ integral. This finally yields

$$\left\langle \phi_s^2 \right\rangle_\xi (t) = \frac{1}{2\pi} \iiint_{-\infty}^{\infty} G(-\omega', \omega) G(\omega', \tilde{\omega}') S(\omega') e^{it(\omega + \tilde{\omega}')} \, d\omega \, d\omega' \, d\tilde{\omega}' . \tag{D.5}$$

As a result of the numerical solution of the Green's function, we want to express the above result in terms of the series representation of the Green's function. By direct substitution, we obtain

$$\left\langle \phi_s^2 \right\rangle_\xi = \frac{1}{2\pi} \iiint_{-\infty}^{\infty} \sum_{n,m} G_n(-\omega') G_m(\omega') \delta(-\omega' - \omega - n\omega_b) \delta(\omega' - \tilde{\omega}' - m\omega_b)$$
$$\times S(\omega') e^{it(\omega + \tilde{\omega}')} \, d\omega \, d\omega' \, d\tilde{\omega}'$$
$$= \frac{1}{2\pi} \int_{-\infty}^{\infty} \sum_{n,m} G_n(-\omega') G_m(\omega') S(\omega') e^{-it(n+m)\omega_b} \, d\omega' . \tag{D.6}$$

As a final step, let us foresightfully time-average the above expression over one period of oscillation of the mode $\phi_b$. Let us consider the Fourier harmonic resonant to the mode, so that the time-average reads

$$\left\langle \phi_s^2 \right\rangle_{\xi,t} = \frac{\omega_b}{2\pi} \frac{1}{2\pi} \int_0^{2\pi/\omega_b} e^{i\omega_b t} \int_{-\infty}^{\infty} \sum_{n,m} G_n(-\omega) G_m(\omega) S(\omega) e^{-it(n+m)\omega_b} \, d\omega \, dt$$
$$= \frac{\omega_b}{2\pi} \frac{1}{2\pi} \int_{-\infty}^{\infty} \sum_{nm} G_n(-\omega) G_m(\omega) S(\omega) \int_0^{2\pi/\omega_b} e^{-it(n+m-1)\omega_b} \, dt \, d\omega , \tag{D.7}$$

where we rearrange the formula such that we obtain the integral expression of the Kronecker delta as given by the time integral. With this, we simplify the result:

$$\left\langle \phi_s^2 \right\rangle_{\xi,t} (A_b, \theta_b) = \frac{1}{2\pi} \int_{-\infty}^{\infty} \sum_{n,m} G_n(-\omega) G_m(\omega) S(\omega) \delta_{m,-n+1} \, d\omega$$
$$= \frac{1}{2\pi} \int_{-\infty}^{\infty} \sum_n G_n(\omega) G_{n-1}^*(\omega) S(\omega) \, d\omega , \tag{D.8}$$

where we use $G_{-n}(-\omega) = G_n^*(\omega)$. We now have a function of $A_b$ and $\theta_b$ describing the average noise pressure on the mode $\phi_b$.

As mentioned in the main text, in order to derive the equations of motion for the amplitude and phase of the mode $\phi_b$, we once again use $\phi_b(t) = A_b(t)\left(e^{-i[\omega_b t + \theta_b(t)]} + e^{i[\omega_b t + \theta_b(t)]}\right)$. The derivatives then read

$$\dot{\theta}_b(t) = \dot{A}_b(t) e^{-i[\omega_b t + \theta_b(t)]} - iA_b(t)[\omega_b + \dot{\theta}_b(t)] e^{-i[\omega_b t + \theta_b(t)]} + \text{c.c.} , \tag{D.9}$$

$$\ddot{\phi}_b(t) = \ddot{A}_b(t) e^{-i[\omega_b t + \theta_b(t)]} - 2i\dot{A}_b(t)[\omega_b + \dot{\theta}_b(t)] e^{-i[\omega_b t + \theta_b(t)]}$$
$$- iA_b(t)\ddot{\theta}_b(t) e^{-i[\omega_b t + \theta_b(t)]} - A_b(t)[\omega_b + \dot{\theta}_b(t)]^2 e^{-i[\omega_b t + \theta_b(t)]} + \text{c.c.} \tag{D.10}$$

In the spirit of the WKB approximation, let us immediately drop the second derivatives, products and powers of derivatives, and the terms $\gamma_b \dot{A}_b$ and $\gamma_b \dot{\theta}_b$. Upon inserting the Fourier component shown above of the remaining expressions into Eq. (24), we obtain

$$\left[-2i\omega_b \dot{A}_b(t) - 2\omega_b A_b(t)\dot{\theta}_b(t) - 2i\gamma_b A_b \omega_b\right] e^{-i[\omega_b t + \theta_b(t)]} - g_b^2 \left\langle \phi_s^2 \right\rangle_\xi (A_b, \theta_b) = 0 . \tag{D.11}$$

The reason for computing the time-average above now becomes apparent: multiplying this equation by $e^{i[\omega_b t + \theta_b(t)]}$ and time-averaging over the whole equation gives the exact term computed above multiplied by $e^{i\theta_b(t)}$. Decomposing this result into the real and imaginary parts gives

$$A_b(t)\dot{\theta}_b + \frac{g_g^2}{2\omega_b}\text{Re}[\langle\phi_s^2\rangle_{\xi,t}(A_b,\theta_b)e^{i\theta_b(t)}] = 0,\tag{D.12}$$

$$\dot{A}_b(t) + \gamma_b A_b + \frac{g_g^2}{2\omega_b}\text{Im}[\langle\phi_s^2\rangle_{\xi,t}(A_b,\theta_b)e^{i\theta_b(t)}] = 0.\tag{D.13}$$

In the expression given in the main text Eq. (27) the phase exponent $e^{i\theta_b(t)}$ has been included in the definition of $\langle\phi_s^2\rangle_{\xi,t}(A_b,\theta_b)$.

# E  Estimating efficiency

As explained in the main text, in order to estimate the efficiency of the device, we need to compute the net heat flow through the system. We start with the following set of equations describing the stationary system, where we ignore resonator B and threat resonator A and the SQUID loop as one tunable frequency resonator with frequency $\omega_a'$ and field variable $\phi_a'$:

$$\ddot{\phi}_a' + \omega_a'^2\phi_a' + \sum_{f\in[h,c]}\alpha_{fa}\ddot{\phi}_f = 0,\tag{E.1a}$$

$$\ddot{\phi}_f + \omega_f^2\phi_f + 2\gamma_f\dot{\phi}_f + \alpha_f\ddot{\phi}_a' = \xi_f(t).\tag{E.1b}$$

In order to compute the derivative of total energy, we multiply the first equation by $\dot{\phi}_a$ and the latter by $\dot{\phi}_f$. By extracting total time derivatives where possible, we obtain

$$\frac{d}{dt}\left[\frac{\dot{\phi}_a'^2}{2} + \frac{\omega_a'^2}{2}\phi_a'^2\right] + \sum_{f\in[h,c]}\alpha_{fa}\ddot{\phi}_f\dot{\phi}_a' = 0,\tag{E.2a}$$

$$\frac{\alpha_{fa}}{\alpha_f}\frac{d}{dt}\left[\frac{\dot{\phi}_f^2}{2} + \frac{\omega_f^2}{2}\phi_f^2\right] + 2\gamma_f\frac{\alpha_{fa}}{\alpha_f}\dot{\phi}_f^2 + \alpha_{fa}\ddot{\phi}_a'\dot{\phi}_f = \frac{\alpha_{fa}}{\alpha_f}\xi_f(t)\dot{\phi}_f,\tag{E.2b}$$

where we also multiply the latter equation by $\alpha_{fa}/\alpha_f$. Next, we sum together the equations, which yields

$$\frac{d}{dt}\left[\frac{\dot{\phi}_a^2}{2} + \frac{\omega_a'^2}{2}\phi_a'^2 + \sum_{f\in[h,c]}\frac{\alpha_{fa}}{\alpha_f}\left(\frac{\dot{\phi}_f^2}{2} + \frac{\omega_f^2}{2}\phi_f^2 + \alpha_f\dot{\phi}_f\dot{\phi}_a\right)\right] = \sum_{f\in[h,c]}\frac{\alpha_{fa}}{\alpha_f}\left(\xi_f(t)\dot{\phi}_f - 2\gamma_f\dot{\phi}_f^2\right).\tag{E.3}$$

On the left, we identify the time derivative of the total energy of the system. Since all the energy within the system is thermal energy, we can equate this to the derivative of heat, i.e., heat flow through the system. Therefore, we have

$$\dot{Q} = \sum_{f\in[h,c]}\frac{\alpha_{fa}}{\alpha_f}\left(\xi_f(t)\dot{\phi}_f - 2\gamma_f\dot{\phi}_f^2\right).\tag{E.4}$$

The above result is given in terms of the noise function $\xi_f(t)$, and therefore noisy itself. Consequently, we aim to estimate the average heat flow, quite similarly to that for the field $\phi_s$

above. In order to proceed, we need to know the field $\phi_f$. In the stationary case, it can be solved from the above set of equations (E.1) via Fourier transformation. In Fourier space, we have

$$-\omega^2\phi_{\rm a}' + \omega_{\rm a}'^2\phi_{\rm a}' - \omega^2\sum_{f\in[{\rm h,c}]}\alpha_{f{\rm a}}\phi_f = 0, \tag{E.5a}$$

$$-\omega^2\phi_f + \omega_f^2\phi_f - 2{\rm i}\gamma_f\omega\phi_f - \omega^2\alpha_f\phi_{\rm a}' = \xi_f(\omega), \tag{E.5b}$$

which can be solved by standard algebra. For the filter degrees of freedom, we obtain

$$\varphi_f(\omega) = \sum_{f'\in[{\rm h,c}]}G_{ff'}(\omega)\xi_{f'}(\omega), \tag{E.6}$$

with the Green's functions defined as

$$G_{ff}(\omega) = \frac{\mathcal{P}_{f'}(\omega)}{\mathcal{D}(\omega)}, \qquad G_{f\neq f'}(\omega) = \frac{\alpha_f\alpha_{f'{\rm a}}\omega^4}{\mathcal{D}(\omega)}, \tag{E.7}$$

where $f, f' \in [{\rm h,c}]$, and we define

$$\mathcal{P}_f(\omega) = \vartheta_f(\omega)\left(\omega_{\rm a}'^2 - \omega^2\right) - \alpha_f\alpha_{f{\rm a}}\omega^4, \tag{E.8}$$

$$\mathcal{D}(\omega) = \left(\omega_{\rm a}'^2 - \omega^2\right)\prod_{f\in[{\rm h,c}]}\vartheta_f(\omega) - \omega^4\sum_{f\neq f'}\alpha_f\alpha_{f{\rm a}}\vartheta_{f'}(\omega), \tag{E.9}$$

$$\vartheta_f(\omega) = \omega_f^2 - \omega^2 - 2{\rm i}\omega\gamma_f. \tag{E.10}$$

Though we will continue to work in the Fourier domain, we give the solution to the field degrees of freedom in the time domain for completeness, which reads

$$\varphi_f(t) = \sum_{f'\in[{\rm h,c}]}\int_{-\infty}^{\infty}G_{ff'}(t-\tau)\xi_{f'}(\tau)\,{\rm d}\tau. \tag{E.11}$$

It is now possible to estimate the expectation value

$$\langle\dot{Q}\rangle = \sum_{f\in[{\rm h,c}]}\frac{\alpha_{f{\rm a}}}{\alpha_f}\left(\langle\xi_f(t)\dot{\phi}_f\rangle - 2\gamma_f\langle\dot{\phi}_f^2\rangle\right), \tag{E.12}$$

which we can write via Fourier transform as

$$\langle\dot{Q}\rangle = \sum_{f\in[{\rm h,c}]}\frac{\alpha_{f{\rm a}}}{\alpha_f}\left(\frac{1}{4{\rm i}\pi^2}\iint_{-\infty}^{\infty}\omega\langle\xi_f(\omega)\hat{\phi}_f(\omega')\rangle{\rm e}^{-{\rm i}t(\omega+\omega')}\,{\rm d}\omega\,{\rm d}\omega'\right.$$
$$\left. + \frac{\gamma_f}{2\pi^2}\iint_{-\infty}^{\infty}\omega\omega'\langle\hat{\phi}_f(\omega)\hat{\phi}_f(\omega')\rangle{\rm e}^{-{\rm i}t(\omega+\omega')}\,{\rm d}\omega\,{\rm d}\omega'\right). \tag{E.13}$$

The expectation values in the above expression are now computed similarly as above in Sec. D, by using $\langle\xi_f(\omega)\xi_f(\omega')\rangle = 2\pi\delta(\omega+\omega')S_f(\omega)$, and the frequency domain result Eq. (E.6). This quickly yields

$$\langle\dot{Q}\rangle = \sum_{f\in[{\rm h,c}]}\frac{\alpha_{f{\rm a}}}{\alpha_f}\left(\frac{1}{2{\rm i}\pi}\int_{-\infty}^{\infty}\omega G_{ff}^*(\omega)S_f(\omega)\,{\rm d}\omega\right.$$
$$\left. + \frac{\gamma_f}{\pi}\int_{-\infty}^{\infty}\omega^2\left\{|G_{ff}(\omega)|^2 S_f(\omega) + |G_{f\neq f'}(\omega)|^2 S_{f'}(\omega)\right\}\,{\rm d}\omega\right), \tag{E.14}$$

which can be evaluated numerically. The efficiency of the device is now defined, as given by Eq. (30) in the main text.

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
