# Peer review of "Proposal for an autonomous quantum heat engine"

_SciPost Physics, doi:SciPost Phys. 19, 101 (2025)_

## Round 2 · Referee Report · Anonymous (Referee 2) · 2025-6-12

Strengths

  • Plausible experimental feasibility

Weaknesses

  • Unclear discussion of the quantum vs. classical effects in the system

Report

The manuscript discusses theoretical analysis of a superconducting device, and shows that the model predicts microwave generation powered by temperature difference between two resistors. In previous literature, a number of related autonomous heat engines constructed from oscillators and oscillator nonlinearity, and work cycles based on modifying resonator frequency have been discussed theoretically. Here, the main motivation, and the main results appear to provide a detailed suggestion and modeling how to realize such ideas in a superconducting circuit platform.

Technically the work contains a device proposal and the solution of classical equations of motion of a superconducting circuit, driven by noise from resistors. This appears fine in itself, and probably is reasonable model for the proposed system at temperatures high enough. However, I have some comments below which should be clarified.

Although a classical model is not really a "single quantum" heat engine, for realizing the system it's regardless useful to understand the classical behavior. In the manuscript it is also suggested with some justifications that the principle would scale to low temperatures.

As such, the work is probably a useful step toward realizing this type of heat engines on cQED platform, and may be suitable for publication after open questions are clarified.

Requested changes

1 - In Eq. (8) and below symmetrized correlator is used, so in a sense emission and absorption are summed together. If they are different, omega > T considered in the manuscript, one could expect it may have some importance for the heat engine physics studied here.

Implications on the difference to fully classical limit is discussed in Sec. 4.3., mostly stating that the numerical results change quantitatively, and in the introduction noting that linear equations are same in quantum case. Can the final results be understood in terms of quanta emitted from the hot reservoir and absorbed in the cold or in the driven resonator? Do the equations obtained reflect such physics? The exponential dependence in Fig. 4(cd) at low T may be reasonable, but this is found from numerical results so the origin is not unambiguous. I don't find the discussion of the semiclassical approximation very clear in the present version.

2 - The Otto cycle as discussed in Sec. 4.4 is somewhat hard to see and understand from Fig. 8. From the discussion, I would guess what is implied is that (x,y)=(omega_a', <phi^2>) forms a loop with nonzero interior. This could be contrasted to proposals [45] and others, after converting between field amplitude and photon counts.

3 - The negative dissipation occurs only at nonzero resonator amplitude when there is internal dissipation, unless quality factor is very high. What can be said about the threshold where the resonator B starts to oscillate if it starts at rest?

4 - The results are calculated by time-scale separation arguments. Are these assumptions self-consistent with the results for the parameters used, i.e do the slow equations produce slow dynamics? This is not explicitly discussed in the manuscript, but probably would be useful to note it.

Recommendation

Ask for minor revision

  • validity: -
  • significance: -
  • originality: -
  • clarity: -
  • formatting: -
  • grammar: -

Author:  Miika Rasola  on 2025-08-20  [id 5747]

(in reply to Report 2 on 2025-06-12)

General response:

We would like to thank the reviewers for raising valid points and making keen critical
observations about our work. The remarks helped us in improving the manuscript considerably
over the revision process. The common point of the reviews seems to be the
issues raised with the qualitative explanation and intuitive description of the proposed
device, as well as lacking schematic figures supporting such considerations. These issues
have been addressed in the revised manuscript, along with other points raised by the
reviewers. The most important addition is the schematic figure (Fig. 2) added to the
manuscript, much clarifying the intuitive understanding we want to convey. The point-
by-point response for each of the reviews can be found below. Please consult the revised
manuscript to see the updates mentioned within.

In our work, we use quasiclassical Langevin equation derived in Ref. [62] of the
manuscript. The noise source in Eq. (7e) is real-valued, which implies symmetric
form of the correlator in Eq. (8). However, the photon absorption is included via
damping term in Eq. (7e). For example, in case of linear open quantum systems, the
correlation functions obtained by solving quasiclassical Langevin equation can be
proven to be exact. Our system is indeed nonlinear, therefore this formalism is only
approximate. However, we expect it to capture the main features even if \hbar\omega > k_BT

Point-by-point:

1 - Within our quasiclassical formalism, the only quantum feature is the correlation
function of the noise Eq. (8). In the classical limit, \hbar → 0, it reduces to the white Johnson–Nyquist
noise. The fully classical limit discussed in 4.3. is understood in this sense.
To provide more intuitive explanation of the physics of the proposed QHE,
we have added a schematic Fig. 2: the fast resonator A absorbs higher frequency
photons from the hot bath and emits lower frequency photons to the cold bath. The
difference of energy is the produced work which goes to the slow engine mode B.
To connect this cartoon picture with the used formalism, we notice that stochastic
source in the right hand side of Eq. (7e) contains both thermal noise and
quantum zero-point fluctuations (ZPF). If the temperature of hot reservoir is low
k_BT_h ≪ \hbar\omega_h only ZPF part of the noise remain. Therefore, power and efficiency
shown in Fig. 4(cd) is suppressed despite the temperature of cold bath is well below
Th since it is also driven by the ZPF noise. In the photon language, this corresponds
to the vanishing population of the both reservoirs. This picture breaks in classical
limit \hbar → 0 due to the absence of zero-point fluctuations in the noise sources. In
this case, T_c = 0 corresponds to absence of noise and there is always flow of energy
from the hot bath to the cold.

2 - We are grateful to the Referee for this comment. We have changed Fig. 9 of the
manuscript and added the corresponding cycle in the plane (\omega_a', n_a),
where n_a is the mean photon number in the working fluid mode A.
The latter has been estimated as 2E_m/(\hbar\omega_a′), where E_m is the
magnetic energy of the mode A (Eq. (33) of the revised manuscript).
The loop has indeed non-vanishing area and direction of this
loop is such that positive work is produced.

3 - The condition on the quality factor required for the mode B oscillations to start
from zero is even stricter. However, we observe that it is possible in principle: Fig. 4
shows that in case Q = inf we have negative dissipation even at zero amplitude.
To illustrate this better, we have added panels in Figs. 5, 6, and 7 where we show
(i) threshold Q-factor where the resonator B starts to oscillate from zero and (ii)
threshold Q-factor at which self-sustained oscillations are impossible.

4 - Indeed, the fulfilment of this assumption was not explicitly mentioned in the manuscript,
but was only readable from the data presented in Figures. From here we find the
slow dynamics to have a characteristic rate of change less than one per mill of that
of the intermediate dynamics. We have explicitly mentioned this in the updated
manuscript.

---

## Round 2 · Referee Report · Anonymous (Referee 1) · 2025-6-12

Report

The authors present an analysis of a quantum heat engine realized through a superconducting circuit coupled to two LC resonators at different temperatures. Using a quasiclassical model with several approximations, they estimate the engine's efficiency, which is found to be relatively low. While the practical application of this engine may be limited due to its low efficiency, the work holds significant interest from a fundamental perspective. However, the manuscript requires some revision to enhance clarity.

To start with, I miss a more intuitive explanation of the mechanism which drives the engine, i.e. on the origin of the self-sustained oscillations in phi_b. Is there a simple qualitative condition for their appearance? Perhaps a simple schematic figure would help more than the rather involved numerical approach described in the manuscript.

On the other hand, I wonder about the need of such a complex circuit with so many parameters. If this work is motivated by an actual experiméntal device this should be clearly stated. The choice of parameters in table 1 should also be more clearly justified.

Finally, it would be interesting that the authors comment how they plan to approach a complete quantum description of their device.

Recommendation

Ask for minor revision

  • validity: -
  • significance: -
  • originality: -
  • clarity: -
  • formatting: -
  • grammar: -

Author:  Miika Rasola  on 2025-08-20  [id 5745]

(in reply to Report 1 on 2025-06-12)
Category:
answer to question

We would like to thank the reviewers for raising valid points and making keen critical
observations about our work. The remarks helped us in improving the manuscript considerably
over the revision process. The common point of the reviews seems to be the
issues raised with the qualitative explanation and intuitive description of the proposed
device, as well as lacking schematic figures supporting such considerations. These issues
have been addressed in the revised manuscript, along with other points raised by the
reviewers. The most important addition is the schematic figure (Fig. 2) added to the
manuscript, much clarifying the intuitive understanding we want to convey. The point-
by-point response for each of the reviews can be found below. Please consult the revised
manuscript to see the updates mentioned within.

In our manuscript, we originally attempted to avoid extensive phenomenological
discussion since it was presented in a previous work [1] and instead focused more
on a theoretical model which allows one to calculate the output power and the engine
efficiency. However, we agree with the Referee that a qualitative explanation
of QHE operation makes the manuscript more accessible to the readers. Therefore,
we have added a descriptive schematic Figure 2 accompanied by relevant intuitive
description of the device dynamics.

We agree with the Referee that the choice of the circuit was not clearly explained
in the manuscript. Partially, we rely on our previous work [1], cited in the the
manuscript as the source of motivation. However, we also work on the experimental
realization of the proposed quantum heat engine and use our theoretical model
for the design of the actual device and for the analysis of the experimental data.
Therefore, the model cannot be too simplified if we aim for quantitative match
between the theoretical and experimental results. Note that our experiments are
currently on-going and the data is too preliminary to be published at this point.
Thus we limit the scope of the present manuscript to the theoretical analysis. Furthermore,
we would like to point out that the proposed superconducting microwave
circuit is relatively straightforward to fabricate with the current state-of-the-art
methods. The choice of the parameters for the device was simply dictated by experimental
feasibility — the parameters can be realised and the theory shows that
the device can produce enough output power for experimental observation. This
has been more clearly mentioned in the revised manuscript.

Full quantum description of the QHE engine is, in principle, possible using nonlinear
response theory developed in Ref. [2]. This theory allows to describe the dynamics
of the superconducting circuit using a hierarchical equation of motion and evaluate
various correlators of the output field. This approach is technically challenging but
not impossible since HEOM has been proven to be efficient for simulating open
quantum systems in structured environments [3].

[1] Miika Rasola and Mikko Möttönen. Autonomous quantum heat engine based on
non-markovian dynamics of an optomechanical hamiltonian. Scientific Reports,
14(1):9448, Apr 2024.
[2] V. Vadimov, M. Xu, J. T. Stockburger, J. Ankerhold, and M. Möttönen. Nonlinear-
response theory for lossy superconducting quantum circuits. Physical Review Research, 7(1):013317, March 2025.
[3] Meng Xu, J T Stockburger, G Kurizki, and J Ankerhold. Minimal quantum thermal
machine in a bandgap environment: non-markovian features and anti-zeno advantage.
New Journal of Physics, 24(3):035003, March 2022.

---

## Round 4 · Referee Report · Anonymous (Referee 1) · 2025-9-2

Report

The authors have revised their manuscript in accordance to my previous report. The added schematic figure gives a more intuitive idea of their proposal and they have clarified the reason for chosen circuit and parameters values. I would suggest that the authors also give a reference to V. Vadimov et al., Phys. Rev. Res. (2025) as a possible route towards a full quantum treatment.

Recommendation

Publish (easily meets expectations and criteria for this Journal; among top 50%)

---

## Round 4 · Referee Report · Anonymous (Referee 2) · 2025-9-4

Report

The authors have addressed most of my previous comments. The manuscript is probably publishable after the remaining questions below are clarified.

Requested changes

1 - The authors to some degree address the comment on the quantum noise. A remaining technical question is to show there is no significant unphysical energy extraction from ZPF. If the quasiclassical equations were linear, this should work out automatically. As the authors reply, the results (plots) appear to indicate also the model here is OK, but the reason why and to what degree is not clear from the manuscript. Directly commenting on this question would be relevant in the manuscript, given that a heat engine at low temperatures is discussed.

E.g., at zero temperature, Eq. (23) seems to imply phi_s(t)^2 > 0 generally, so phi_b=const. is not exact solution to (24), hence P>0 in (29), although it may be small due to the time scale separation. The dissipation gamma_b is not associated with a Langevin term, so resonator B is damped by a classical T=0 bath, so there should always be energy flow out. Apparently, it is not significant, but some arguments should be given why.

2 - Eq. (22) has some typos, variable n is summation variable in LHS but also appears on RHS.

Recommendation

Ask for minor revision

  • validity: -
  • significance: -
  • originality: -
  • clarity: -
  • formatting: -
  • grammar: -

Author:  Miika Rasola  on 2025-09-25  [id 5863]

(in reply to Report 2 on 2025-09-04)

Comments by Referee #2:

1 - The authors to some degree address the comment on the quantum noise. A remaining technical question is to show there is no significant unphysical energy extraction from ZPF. If the quasiclassical equations were linear, this should work out automatically. As the authors reply, the results (plots) appear to indicate also the model here is OK, but the reason why and to what degree is not clear from the manuscript. Directly commenting on this question would be relevant in the manuscript, given that a heat engine at low temperatures is discussed.

E.g., at zero temperature, Eq. (23) seems to imply phi_s(t)^2 > 0 generally, so phi_b=const. is not exact solution to (24), hence P>0 in (29), although it may be small due to the time scale separation. The dissipation gamma_b is not associated with a Langevin term, so resonator B is damped by a classical T=0 bath, so there should always be energy flow out. Apparently, it is not significant, but some arguments should be given why.

Our response:

Within our model, the energy exchange between the resonators A and B is provided only by performing work on working mode A. We neglect the heat exchange, which is suppressed due to high frequency difference between the modes $\omega_\mathrm b \ll \omega_\mathrm a'$ . Therefore, the model may give zero output even though mode B is treated fully classically without account of quantum noise. We have added a brief clarification in the beginning of Sec. 3.3.
Equation (24) actually allows a zero output power solution $\phi_\mathrm b = const$ which physically corresponds to a dc current in the SQUID. This solution is not quite consistent with the ansatz (18), but we neglect all the off-resonant harmonics in the dynamics of slow mode B, including this one. In Eq. (29), the power is given by amplitude of resonant harmonic at frequency $\omega_\mathrm b$, the possible constant offset is not present in it. We have provided a more detailed discussion on approximations to Eq. (24) in the following paragraph.

Comments by Referee #2:

2 - Eq. (22) has some typos, variable n is summation variable in LHS but also appears on RHS.

Our response:

There is no summation in Eq. (22). The summation sign was a residual typo from earlier version of the Eq. Each equation is labeled by index $n$ as well each component of Green's function. We have removed the summation sign and added $n=0, \pm 1, \pm 2, \ldots$ in the equation to emphasize that it is a numbering, not summation index.

---

## Round 4 · Author Response

We would like to thank the reviewers for raising valid points and making keen critical observations about our work. The remarks helped us in improving the manuscript considerably over the revision process. The common point of the reviews seems to be the issues raised with the qualitative explanation and intuitive description of the proposed device, as well as lacking schematic figures supporting such considerations. These issues have been addressed in the revised manuscript, along with other points raised by the reviewers. The most important addition is the schematic figure (Fig. 2) added to the manuscript, much clarifying the intuitive understanding we want to convey. The point-by-point response for each of the reviews can be found in the comment section. Please consult the revised manuscript to see the updates mentioned within.

---

## Round 4 · List of Changes

Main changes:

  • Improved intuitive discussion and better schematics describing device and the dynamics in the introduction
  • Added analysis about the limiting quality factors of the 'driving mode': when can the QHE start spontaneously vs. when will it stop working altogether
  • Improved comparison with Otto cycle: added figures describing the thermodynamic cycle of the working fluid in (photon number, angular frequency) plane
  • Fixed typos and misleading / confusing sentences

---

## Round 5 · Author Response

Dear Editor,

We are glad to see that Referee #1 is satisfied with our revision of the paper. Hopefully, after addressing the questions raised by Referee #2, our manuscript is suitable for publication in SciPost Physics.

We have addressed both of the comments by referee #2 by providing more detailed discussion on the topics in the revised manuscript and by answering their questions by replying to them in the revision-comments section. Furthermore we haver fixed the error pointed out in Eq. (22).

---

## Round 5 · List of Changes

• We provide more justification to our approximations and detail the assumption and region of validity of our approach more clearly by:
    • Adding a brief clarification in the beginning of Sec. 3.3.
    • Justifying that equation (24) actually allows a zero output power solution
  • We removed the erronous summation sign in Eq. (22)

---

## Editorial Decision

published